# Conditional Coverage Diagnostics for Conformal Prediction

**Sacha Braun** [1 2] **David Holzmüller** [3] **Michael I. Jordan** [4 1] **Francis Bach** [1 2]

## Abstract

Evaluating conditional coverage remains one of the most persistent challenges in assessing the reliability of predictive systems. Although conformal methods can give guarantees on marginal coverage, no method can guarantee to produce sets with correct conditional coverage, leaving practitioners without a clear way to interpret local deviations. To overcome sample-inefficiency and overfitting issues of existing metrics, we cast conditional coverage estimation as a classification problem. Conditional coverage is violated if and only if some classifier can achieve lower risk than the target coverage. Through the choice of a (proper) loss function, the resulting risk difference gives a conservative estimate of natural miscoverage measures such as L1 and L2 distance, and can even separate the effects of over- and under-coverage, as well as handle non-constant target coverages. We call the resulting family of metrics *excess risk of the target coverage* (ERT). We show experimentally that the use of modern classifiers provides much higher statistical power than simple classifiers underlying established metrics like CovGap. Additionally, we use our metric to benchmark different conformal prediction methods. Finally, we release an open-source package for ERT as well as previous conditional coverage metrics. Together, these contributions provide a new lens for understanding, diagnosing, and improving the conditional reliability of predictive systems.

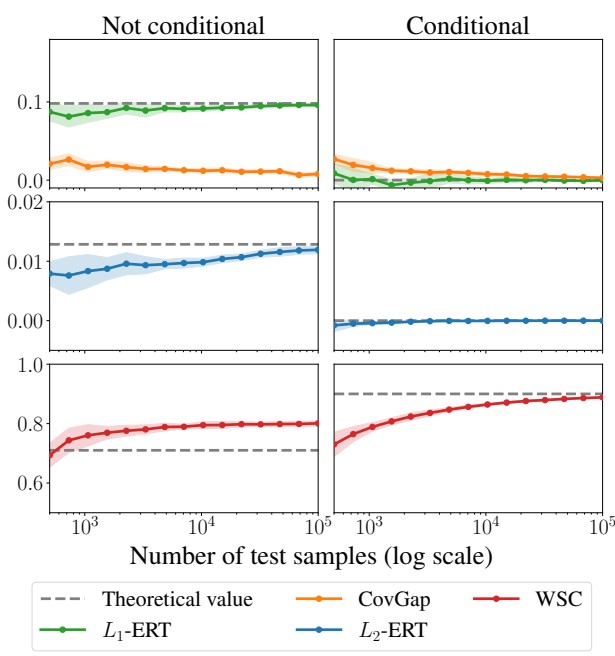

*Figure 1.* Estimated metrics as a function of the number of test samples. Top: CovGap and $L_1$-ERT (which both aim to estimate $\mathbb{E}_X[|\mathbb{P}(Y \in C_\alpha(X)|X) - (1-\alpha)|]$). Middle: $L_2$-ERT. Bottom: WSC. **Left:** Standard CP not conditional. **Right:** Oracle conditional sets. Theoretical values are estimated using the true value $\mathbb{P}(Y \in C_\alpha(X) \mid X)$ with 300,000 samples from $\mathbb{P}_X$.

## 1. Introduction

Uncertainty quantification is central to decision-making across science, engineering, and policy. In many appli-

cations, the goal is not a single point prediction but a set of plausible outcomes with a desired confidence level. This is formalized by a predictive set rule $C(\cdot)$, which outputs a region expected to contain the true outcome with a desired probability. Such predictive sets capture data noise, model imperfections, and variability, supporting safer decisions and clearer communication of model confidence.

Conformal prediction (CP) offers a general framework for constructing prediction sets with finite-sample coverage guarantees (Vovk et al., 2005; Shafer & Vovk, 2008). Its only requirement is that the available data samples are exchangeable: a condition even weaker than the standard independent and identically distributed (i.i.d.) assumption. In recent years, CP has rapidly emerged as a go-to tool for adding rigorous, model-agnostic uncertainty estimates to modern black-box predictors (Angelopoulos & Bates, 2023). This makes it especially appealing for scientific and

[1]Sierra team, Inria Paris, France [2]Ecole Normale Supérieure, PSL Research University, Paris [3]Soda team, Inria Paris-Saclay, France [4]Departments of EECS and Statistics, UC Berkeley, USA. Correspondence to: Sacha Braun <sacha.braun@inria.fr>.

*Proceedings of the 43rd International Conference on Machine Learning*, Seoul, South Korea. PMLR 306, 2026. Copyright 2026 by the author(s).

industrial applications where formal guarantees on model predictions are essential.

CP's simplicity hides an important drawback: it only guarantees *marginal coverage*, which means that the constructed prediction set contains the true outcome with probability $1-\alpha$ on average across the population. That is, the coverage is right on average, but not necessarily for each individual. In practice one often desires *conditional coverage*; i.e., asking that the coverage guarantee holds not only on average but also for specific subpopulations or feature values.

Achieving exact conditional coverage is impossible in general without strong distributional assumptions (Vovk, 2012; Lei & Wasserman, 2014; Foygel Barber et al., 2021), and even approximate versions are notoriously difficult to deploy. Improving conditional coverage in CP typically requires carefully designed nonconformity scores. Common strategies rely on models that provide uncertainty estimates, such as quantile regression (Romano et al., 2019), predictive distributions (Izbicki et al., 2022; Braun et al., 2025b), or local score adjustments (Guan, 2023; Messoudi et al., 2022; Thurin et al., 2025). Recent work proposes post hoc corrections that directly model conditional quantiles of the nonconformity score (Plassier et al., 2025a). There is, however, a difficulty in assessing whether progress is being made in this literature, which is the lack of a standard way to *evaluate* conditional coverage and thereby compare algorithms. The fundamental problem of the evaluation of conditional coverage is the focus of the current paper.

Evaluating conditional coverage is difficult. Group-based diagnostics, such as fairness-style coverage gaps (Ding et al., 2023), require large sample sizes per group and are highly sensitive to group definitions. Geometric scans such as worst-case slab coverage (WSC, Cauchois et al., 2021) offer a more adaptive view but suffer from severe sample complexity in high dimensions. Dependence-based diagnostics (Feldman et al., 2021) capture correlations between coverage and auxiliary variables but do not provide a standalone notion of conditional validity. In short, there is still no robust, general-purpose metric for assessing conditional coverage in practice.

**Contributions.** We address this gap by reframing conditional coverage evaluation as a supervised prediction task: given features $X \in \mathcal{X}$, predict whether the label $Y \in \mathcal{Y}$ falls inside the predictive set $C_\alpha(X)$, where $(X, Y) \sim \mathbb{P}_{X,Y}$. Under perfect conditional coverage, for any proper loss $\ell$ the Bayes-optimal predictor, $h : \mathcal{X} \to [0, 1]$, which is defined as the minimizer of the risk $\mathbb{E}[\ell(h(X), \mathbb{1}\{Y \in C_\alpha(X)\})]$ is the constant $1 - \alpha$. Consequently, any predictor that consistently outperforms this constant directly exposes a violation of conditional coverage. Building on this insight, we introduce the *excess risk of the target coverage* ($\ell$-ERT) metric

to quantify deviations from conditional validity. Our metric provides an estimate of $\mathbb{E}_X[D_\varphi(p(X) \| 1 - \alpha)]$ where $D_\varphi$ is the Bregman divergence of a convex function $\varphi$ (Bregman, 1967). For instance, we can reliably estimate the quantity $\mathbb{E}[|\mathbb{P}(Y \in C_\alpha(X)|X) - (1 - \alpha)|]$ and our estimator is guaranteed to provide a lower bound on its true value.

To contextualize our contribution, we establish a formal connection between existing group-based diagnostics and our metrics, showing that our formulation generalizes partition-based estimators to arbitrary predictor classes. This unified perspective unlocks the full potential of functional estimation, integrating both parametric and nonparametric approaches to assess conditional coverage, moving beyond heuristic group evaluations toward principled, model-based inference.

Through experiments, we demonstrate that our proposed conditional coverage metrics are empirically more robust, that is, less prone to misleading diagnostics, than existing alternatives. Finally, we benchmark several conformal prediction methods on real-world regression and classification tasks to compare their conditional coverage performance. To support reproducibility and to catalyze further progress in conformal prediction, we release `covmetrics`[1] an open-source package for evaluating conditional coverage using our approach alongside established metrics.

Overall, this work casts *conditional coverage evaluation* as a key missing piece for practical conformal prediction, and offers new tools to fill the gap.

**Background on conformal prediction.** We summarize the usual conformal prediction (Papadopoulos et al., 2002; Lei et al., 2018; Angelopoulos & Bates, 2023) procedure in Appendix B. It is a procedure to build a predictive set $C_\alpha(X_{\text{test}})$ such that

$$\mathbb{P}_{X,Y}(Y_{\text{test}} \in C_\alpha(X_{\text{test}})) = 1 - \alpha. \qquad (1)$$

It is important to note that the guarantee in Eq. (1) is marginal, which means that coverage holds on average over the distribution of $X_{\text{test}}$ and a dataset used to build $C_\alpha$. A stronger requirement is *conditional coverage*, which demands

$$\mathbb{P}_{Y|X}(Y_{\text{test}} \in C_\alpha(X_{\text{test}}) \,|\, X_{\text{test}}) = 1 - \alpha, \qquad (2)$$

for almost every $X_{\text{test}}$, but achieving (2) is impossible in general without additional assumptions. For a given conformal prediction strategy that achieves marginal coverage (1), it is essential to be able to measure how close to a conditional guarantee (2) we are.

**Notation.** We denote by $\mathbb{1}_{x \in A}$ the indicator function, equal to 1 if $x \in A$ and to 0 if $x \notin A$, for some

---

[1] https://github.com/ElSacho/covmetrics

set $A$. We denote by $\mathrm{sgn}(x)$ the function that returns the sign of $x \in \mathbb{R}$, where $\mathrm{sgn}(0) = 0$. We write $\Delta_d := \left\{ p \in \mathbb{R}^d \mid p_i \geq 0 \; \forall i = 1, \ldots, d, \; \sum_{i=1}^{d} p_i = 1 \right\}$ to denote the probability simplex.

## 2. Related Work

In the following, we assume that a predictive model has already been trained to produce a predictive set rule $C_\alpha(\cdot)$ for the output variable $Y \in \mathcal{Y}$, given a feature vector $X \in \mathcal{X}$. We are given a test dataset, $\mathcal{D}_{\text{test}} = \{(X_i, Y_i)\}_{i=1}^{m}$, to evaluate the conditional coverage of this predictive strategy. The test samples are assumed to be sampled i.i.d. from the distribution $\mathbb{P}_{X,Y}$ and unseen during training.

**Group-based diagnostics.** A common way to study conditional coverage for a predictive set $C_\alpha(\cdot)$ is to evaluate coverage over subpopulations or other partitions of the data. To make this concrete, fix a finite set of groups $\mathcal{G}$ and a mapping $g : \mathcal{X} \to \mathcal{G}$ that assigns each feature $x \in \mathcal{X}$, to a group $g(x) \in \mathcal{G}$. For a given group $\mathbf{g} \in \mathcal{G}$ and associated test indices $\mathcal{I}_{\mathbf{g}} = \{i : g(X_i) = \mathbf{g}\}$, the empirical coverage in group $\mathbf{g}$ is

$$C_{\mathbf{g}} = \frac{1}{|\mathcal{I}_{\mathbf{g}}|} \sum_{i \in \mathcal{I}_{\mathbf{g}}} \mathbb{1}\{Y_i \in C_\alpha(X_i)\}.$$

We review several strategies for defining these groups in Appendix A. Unless stated otherwise, in the following, the groups are obtained by clustering the feature space using the k-means algorithm. Below we review strategies that use such groupings to diagnose conditional miscoverage.

- **Coverage gap (CovGap).** This measures the average absolute deviation from the target coverage across groups,

$$\text{CovGap} = \frac{1}{|\mathcal{G}|} \sum_{\mathbf{g} \in \mathcal{G}} \big| C_{\mathbf{g}} - (1 - \alpha) \big|.$$

  This metric is one of the most commonly used metrics in the literature (see, e.g, Ding et al. 2023; Kaur et al. 2025; Zhu et al. 2025; Fillioux et al. 2026; Liu et al. 2026). A problem is that CovGap requires a large number of samples within each group to be consistent.

- **Weighted coverage gap (WCovGap).** To connect CovGap to our main metrics, we introduce its corrected version that assigns a weight to each group's coverage gap:

$$\text{WCovGap} = \sum_{\mathbf{g} \in \mathcal{G}} \frac{|\mathcal{I}_{\mathbf{g}}|}{m} \big| C_{\mathbf{g}} - (1 - \alpha) \big|.$$

  This formulation highlights that this metric can be interpreted as a nonparametric estimator of the quantity,

$$\mathbb{E}_X \big[ \big| \mathbb{P}_{Y|X}(Y \in C_\alpha(X) \mid X) - (1 - \alpha) \big| \big].$$

Indeed, $\text{CovGap}(X) := C_{g(X)}$ is an estimate of $\mathbb{P}(Y \in C_\alpha(X) \mid g(X) = \mathbf{g})$, and $\frac{|\mathcal{I}_{\mathbf{g}}|}{m}$ an estimate of $\mathbb{P}_X(g(X) = \mathbf{g})$. Under standard regularity assumptions, specifically, if the partition $\mathcal{G}$ becomes increasingly fine (i.e., the number of groups tends to infinity while their diameters shrink to zero) and if $h^*(X) := \mathbb{E}[\mathbb{1}_{Y \in C_\alpha(X)} \mid X]$ is Lipschitz-continuous, then the groupwise coverage $C_{\mathbf{g}}$ converges to the true conditional coverage $\mathbb{P}_{Y|X}(Y \in C_\alpha(X) \mid X)$ (see, e.g, Györfi et al. 2002; Bach 2024). Consequently, the weighted CovGap (WCovGap) metric provides a nonparametric estimate of the stated quantity. If the groups are balanced in size, the metric CovGap admits the same type of probabilistic interpretation.

This observation is central to understanding the positioning of our work: previous strategies can be seen as partition-wise estimators of conditional coverage. In contrast, our approach leverages modern classifiers to obtain a more accurate estimation of conditional coverage.

**Worst-case slab diagnostic.** Rather than pre-specified groups, some diagnostics scan geometric slices of the feature space, if $\mathcal{X} \subset \mathbb{R}^d$. This strategy is commonly referred to as the worst-case slab coverage (WSC, Cauchois et al., 2021).

- **Worst-case slab coverage (WSC).** For a direction $v \in \mathbb{R}^d$ and scalars $a < b$, define the slab

$$S_{v,a,b} := \{x \in \mathbb{R}^d : a \leq v^\top x \leq b\}.$$

Let $\mathcal{I}_{v,a,b} = \{i : X_i \in S_{v,a,b}\}$. For a mass threshold $\delta \in (0, 1]$, the empirical WSC in direction $v$ $\text{WSC}_n\big(C_\alpha(\cdot), v\big)$ is

$$\inf_{a<b} \left\{ \frac{1}{|\mathcal{I}_{v,a,b}|} \sum_{i \in \mathcal{I}_{v,a,b}} \mathbb{1}\{Y_i \in C_\alpha(X_i)\} \; \middle| \; \frac{|\mathcal{I}_{v,a,b}|}{n} \geq \delta \right\}.$$

In practice, the induced metric requires a finite set of directions $V$ and computes:

$$\text{WSC} = \inf_{v \in V} \text{WSC}_n\big(C_\alpha(\cdot), v\big).$$

This set is typically generated by sampling vectors at random from $\mathbb{R}^d$. When evaluating over a finite set of directions $V$, WSC uniformly approximates the population slab-coverage with high probability; the approximation error depends on the VC dimension (Vapnik, 2000) of the class of slabs induced by $V$. However, under conditional coverage, without sufficient test data WSC tends to provide a pessimistic estimate of the conditional coverage violation by overstating apparent conditional coverage violations as detailed in Appendix A. Furthermore, this strategy does not adapt well to categorical data.

In Appendix A we review additional metrics. One of them is feature-stratified coverage (FSC), a group-based metric that reports the group with the worst coverage. This metric often appears in fairness-related work (see e.g, Angelopoulos & Bates 2023; Ding et al. 2023; Jung et al. 2023). Several grouping strategies besides categorical attributes or clustering have also been studied. For example, equal opportunity of coverage (EOC) (Wang et al., 2023a) forms groups based on the output, and size-stratified coverage (SSC) (Angelopoulos et al., 2021) groups examples by the size of the prediction set. Another approach is to avoid explicit grouping and instead measure statistical dependence between the coverage indicator $Z := \mathbf{1}\{Y \in C_\alpha(X)\}$ and the prediction-set size. Feldman et al. (2021) introduced two such dependence measures based on Pearson's correlation and the Hilbert–Schmidt independence criterion (HSIC).

Each diagnostic has strengths and limitations. Group-based metrics (CovGap, FSC, EOC, SSC) are intuitive and directly tied to fairness-style guarantees, but their statistical power depends strongly on the choice of groups and on having enough data per group. Geometric scans like WSC explore slices of $\mathcal{X}$ without pre-specified semantic groups but suffer from the complexity of the feature space. Representation-based measures (Pearson, HSIC) provide complementary, model-driven checks for dependence between coverage and auxiliary signals, but low dependence does not prove full conditional coverage. A central challenge in catalyzing research progress in conformal prediction is the lack of reliable ways to assess conditional coverage empirically. Although many recent methods are designed to improve conditional coverage (Gibbs et al., 2025; Ding et al., 2023; Kaur et al., 2025; Plassier et al., 2025b), existing guarantees are largely theoretical, and robust practical metrics remain elusive.

## 3. Evaluating Conditional Coverage

We would like the conditional coverage

$$p(x) := \mathbb{P}(Y \in C_\alpha(X) \mid X = x)$$

to be equal to $1-\alpha$ $\mathbb{P}_X$-almost surely. Introducing the binary random variable $Z = \mathbb{1}\{Y \in C_\alpha(X)\}$, we can rewrite

$$p(x) = \mathbb{P}(Z = 1|X = x).$$

Estimating $\mathbb{P}(Z = 1|X = x)$ is a binary classification problem, as we have access to a dataset of pairs $(X_i, Z_i)$. Some metrics such as CovGap or WSC implicitly learn classifiers based on histograms or slabs. However, these methods are rarely used for classification due to their poor practical performance. Explicitly reformulating conditional miscoverage estimation as a classification problem allows us to leverage strong and practically proven classifiers. Once a classifier $h : \mathcal{X} \to [0,1]$ is trained, we still need to use it to assess conditional miscoverage. The key idea is that under conditional coverage, given a proper score $\ell$, no classifier can achieve a lower risk than the constant predictor $1 - \alpha$. If we can learn a predictor that performs better, then conditional coverage does not hold. This leads to a metric with theoretical guarantees and a clear interpretation. In particular, our metric is a conservative estimate of $\mathbb{E}[d(1 - \alpha, p(X))]$ for any $d : [0,1] \times [0,1] \to \mathbb{R}$ such that for all $p \in [0,1]$, $d(p, \cdot)$ is convex and minimized at $p$.

### 3.1. Excess risk of the target coverage (ERT)

For a given classifier $h$ and loss function $\ell$, the associated risk is defined as

$$\mathcal{R}_\ell(h) := \mathbb{E}_{X,Z}[\ell(h(X), Z)].$$

The Bayes predictor in this task is (see, e.g., Devroye et al. 2013):

$$h^*(x) \in \underset{q \in [0,1]}{\operatorname{argmin}} \, \mathbb{E}[\ell(q, Z) \mid X = x].$$

If the loss $\ell$ is a proper loss (see, e.g, Gneiting & Raftery 2007; Bröcker 2009), then it is optimal to predict the true probability $h^*(X) = \mathbb{E}[Z|X] = p(X)$ $\mathbb{P}_X$-almost surely. If conditional coverage holds, we get $h^*(X) = \mathbb{E}[Z|X] = 1-\alpha$ $\mathbb{P}_X$-almost surely, so no classifier can achieve lower risk than the constant $1-\alpha$ prediction. Proper losses include the Brier score, $\ell(p, y) = (p - y)^2$, and the log-loss score, $\ell(p, y) = -y \log p - (1 - y) \log(1 - p)$, where $p \in [0, 1]$ denotes the predicted probability of the event occurring and $y \in \{0, 1\}$ denotes the observed outcome.

This motivates the *excess risk of the target coverage* ($\ell$-ERT). For a general proper loss $\ell$, we define

$$\ell\text{-ERT} := \mathcal{R}_\ell(1 - \alpha) - \mathcal{R}_\ell(p).$$

Larger values of $\ell$-ERT correspond to greater violations of conditional coverage.

**Interpretation and examples.** ERT has a probabilistic interpretation. Indeed,

$$\ell\text{-ERT} = \mathbb{E}_X[d_\ell(1 - \alpha, p(X))],$$

where $d_\ell(p, q) := \mathbb{E}_{y \sim q}[\ell(p, y) - \ell(q, y)]$ is the divergence associated with the proper score $\ell$ (see, e.g, Bröcker 2009). A justification of this equality is provided in Appendix C. We summarize the $\ell$-ERT scores for different proper scores in Table 1.

We will show in Section 3.2 that a general class of convex distances can be estimated via ERTs. This allows us to define a proper score for $L_1$-ERT, that directly targets the estimation of the quantity $\mathbb{E}_X[|\mathbb{P}(Y \in C(X)|X) - (1-\alpha)|]$.

*Table 1.* Examples of proper scoring rules and their associated ERT scores.

| Name | Proper score $\ell(p, y)$ | $\ell$-ERT formula |
|------|---------------------------|--------------------|
| $L_1$-ERT | $\mathrm{sgn}(p - (1 - \alpha))(1 - \alpha - y)$ | $\mathbb{E}_X[|p(X) - (1 - \alpha)|]$ |
| $L_2$-ERT | $(y - p)^2$ | $\mathbb{E}_X[(1 - \alpha - p(X))^2]$ |
| KL-ERT | $-\log p_y$ | $\mathbb{E}_X[D_{\mathrm{KL}}(p(X) \| 1 - \alpha)]$ |

**Estimation from finite samples.** Since $p(X)$ is unknown in practice, we define the functional

$$\ell\text{-ERT}(h) := \mathcal{R}_\ell(1 - \alpha) - \mathcal{R}_\ell(h), \quad (3)$$

This metric quantifies how much better a predictor $h$ performs relative to the constant baseline $1 - \alpha$. While we cannot guarantee that the learned predictor $h$ coincides with the Bayes-optimal predictor $h^*$, our procedure always provides a lower bound on the true $\ell$-ERT: for all measurable classifiers,

$$\ell\text{-ERT}(h) \leq \ell\text{-ERT}.$$

Therefore, it suffices to find an $h$ that performs better than the constant $1 - \alpha$ to conclude that conditional coverage is not achieved, and use $\ell$-ERT$(h)$ to lower-bound $\mathbb{E}[d_\ell(1 - \alpha, p(X))]$. This probabilistic perspective makes the metric highly interpretable when assessing coverage deviations.

To estimate $\ell$-ERT$(h)$, we evaluate the empirical risk

$$\widehat{\ell\text{-ERT}}(h) := \frac{1}{m} \sum_{i=1}^{m} \left[ \ell(1 - \alpha, Z_i) - \ell(h(X_i), Z_i) \right].$$

To avoid overfitting and misleading diagnostics, we cannot train $h$ on the values $X_i$ that it is evaluated on. In general, cross-validation can be used for this purpose, where multiple classifiers are trained on different subsets of the data, such that each data point can be evaluated using a classifier that was not trained on it. For random forest, we can use out-of-bag predictions. The resulting algorithm for evaluating conditional coverage using $k$-fold cross-validation is summarized in Algorithm 1.

Figure 2 illustrates the usefulness of our metric by showing prediction sets produced under different conformal strategies together with their estimated conditional coverage. The function $h$ is estimated with a neural network that has two hidden layers of width 64. In the first strategy, which applies a non conditional conformal method, the prediction sets fail to reflect local variations in conditional miscoverage. This leads to large estimated ERT values, with $L_1$-ERT$(h) \approx 0.0757$ and $L_2$-ERT$(h) \approx 0.0073$. In the second strategy, where prediction sets are closer to satisfying conditional coverage, the estimator $h$ estimates coverage levels near $1 - \alpha$. The resulting ERT values are closer to zero,

---

**Algorithm 1** Compute $\widehat{\ell\text{-ERT}}$.

1: **Input:** Data $\{(X_i, Z_i)\}_{i=1}^{m}$, number of folds $k \geq 2$, proper score $\ell$, level $\alpha$, classification method.
2: **Partition the data:** Randomly divide $\{1, \dots, m\}$ into $k$ approx. equal-sized folds $\{\mathcal{I}_1, \dots, \mathcal{I}_k\}$.
3: **for** $j = 1$ to $k$ **do**
4:     **Define folds:** $\mathcal{I}_{\mathrm{val}}^{(j)} = \mathcal{I}_j$,    $\mathcal{I}_{\mathrm{tr}}^{(j)} = \{1, \dots, m\} \setminus \mathcal{I}_j$.
5:     **Train classifier:** Fit a classifier $h^{(j)}$ on $\{(X_i, Z_i) \mid i \in \mathcal{I}_{\mathrm{tr}}^{(j)}\}$ using the specified method.
6:     **Evaluate on validation fold:** For each $i \in \mathcal{I}_{\mathrm{val}}^{(j)}$, compute

$$\widehat{\ell\text{-ERT}}^{(j)} = \frac{1}{|\mathcal{I}_{\mathrm{val}}^{(j)}|} \sum_{i \in \mathcal{I}_{\mathrm{val}}^{(j)}} \ell(1 - \alpha, Z_i) - \ell(h^{(j)}(X_i), Z_i).$$

7: **end for**

8: **Aggregate across folds:** $\widehat{\ell\text{-ERT}} = \sum_{j=1}^{k} \frac{|\mathcal{I}_j|}{m} \widehat{\ell\text{-ERT}}^{(j)}$.

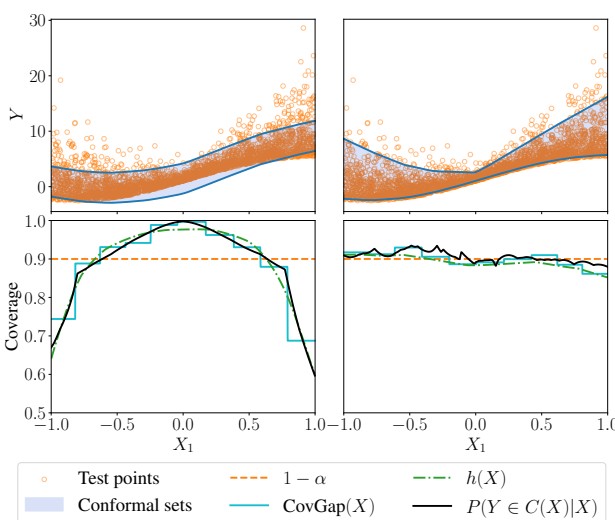

*Figure 2.* Illustration of conditional coverage estimation. The top panel shows data generated from $Y \sim \mathcal{N}(f(X), \sigma(X))$, $X \sim \mathcal{U}([-1, 1])$ with $f(x) = 3\sin(x) + e^x$ and $\sigma(x) = 1/2 + |x| + x^2$, and their predictive sets. The bottom panel shows the conditional coverage estimation $h$ used to estimate the $L_2$-ERT$(h)$, conditional coverage estimation induced by a partition-wise estimator, true conditional coverage, and desired $1 - \alpha$ conditional coverage. **Left:** Conformal sets from the score $S(X, Y) = |Y - \hat{f}(X)|$. **Right:** Conformal sets by fitting quantiles $\alpha/2$ and $1 - \alpha/2$ of $\mathbb{P}_{Y|X}$ following the procedure in Romano et al. (2019).

with $L_1$-ERT$(h) \approx 0.0148$ and $L_2$-ERT$(h) \approx -0.00002$, which signals improved conditional behavior.

We also compare our functional estimator $h$ to the partition-based nonparametric estimator $\mathrm{CovGap}(X)$. In one dimension, the feature space is simple to cluster and the partition-based approach can approximate $p$ well. This advantage does not persist as the feature dimension grows, a point that

will be demonstrated in Section 4.

## 3.2. Estimating general distances

Previously, Table 1 illustrated that specific choices of the proper loss $\ell$ can recover common distance functions. However, we can go much beyond that and estimate any convex distance function $f(q) = d(1 - \alpha, q)$ using an ERT, as long as the proper score $\ell$ is allowed to depend on $f$ and therefore the coverage $1 - \alpha$ itself. The following proposition formalizes this statement.

**Proposition 3.1** (Representing convex losses as ERTs). *Let $f : [0, 1] \to \mathbb{R}_{\geq 0}$ be convex with $f(1 - \alpha) = 0$. Let $f'$ be a subderivative of $f$ satisfying $f'(1 - \alpha) = 0$. Then, the function with ($p \in [0, 1], y \in \{0, 1\}$)*

$$\ell(p, y) := \ell_{f, f'}(p, y) := -f(p) - (y - p)f'(p)$$

*is a proper score satisfying*

$$\ell\text{-ERT} = \mathbb{E}_X[f(p(X))] .$$

A related formulation in Appendix C shows that ERT can estimate $d$ if it is a Bregman divergence of convex functions.

## 3.3. Separating over-coverage and under-coverage

Proposition 3.1 implies that we can estimate asymmetric distance measures to gain more insights on the nature of miscoverage. In particular, one can decompose the convex function $f$ from above as $f = f_+ + f_-$ with an over-coverage part $f_+(p) := f(\max\{p, 1 - \alpha\})$ that only penalizes the case $p > 1 - \alpha$ and an under-coverage part $f_-(p) = f(\min\{p, 1 - \alpha\})$ that only penalizes $p < 1 - \alpha$. Correspondingly, one can decompose the proper loss $\ell$ and the ERT as

$$\ell(p, y) = \ell_+(p, y) + \ell_-(p, y) - \ell(1 - \alpha, y)$$
$$\ell\text{-ERT} = \ell_+\text{-ERT} + \ell_-\text{-ERT}$$
$$\ell_+(p, y) := \ell(\max\{p, 1 - \alpha\}, y)$$
$$\ell_-(p, y) := \ell(\min\{p, 1 - \alpha\}, y) .$$

Together, $\ell_+$-ERT and $\ell_-$-ERT provide a decomposition of conditional coverage error into two complementary components. The first identifies unnecessary conservatism, while the second highlights locations where $C_\alpha$ is too aggressive and exhibits under-coverage. This split view delivers more informative diagnostics and supports targeted improvements in the design of conformal prediction methods. Further experiments to illustrate those diagnostics are presented in Appendix I.

## 3.4. Extensions

**A proxy for conditional coverage.** The learned predictor $h$ can also be used as a proxy for conditional coverage.

For a given test point $X_{\text{test}}$, its conditional coverage can be estimated as

$$h(X_{\text{test}}) \approx \mathbb{P}\big(Y_{\text{test}} \in C_\alpha(X_{\text{test}}) \mid X_{\text{test}}\big).$$

This approximation provides a corrective proxy for conditional coverage: rather than modifying the predictive set $C_\alpha(X_{\text{test}})$, we adjust its predicted coverage level from $1 - \alpha$ to $h(X_{\text{test}})$.

**Evaluating conditional coverage rules.** We further extend our metric to settings where the target conditional coverage is not fixed at $1 - \alpha$, but instead varies according to a specified decision rule. This extension, detailed in Appendix E, enables testing whether a given strategy satisfies conditional coverage with respect to adaptive or context-dependent coverage levels.

# 4. Experiments

Our code is accessible and reproducible from the GitHub repositories[2]. Additional experiments, including a benchmark of the conditional coverage properties of different conformal prediction strategies is provided in Appendix G.

## 4.1. Comparing different classifiers

The quality of the ERT estimation hinges on the choice of a good classifier, which depends on the data type of $X$. We will restrict our experiments to the case where $X$ is a fixed-dimensional vector of numerical and/or categorical features, also known as *tabular data*. For other modalities like images or text, tabular classifiers could be used on top of embeddings of $X$ to keep the training fast. As we want our metric to be reasonably fast to compute, we are particularly interested in finding fast classifiers. For this reason, we do not tune the hyperparameters of the classifiers. Based on recent benchmarks (Erickson et al., 2025; Holzmüller et al., 2024), we choose a subselection of classifiers that are promising in terms of their speed-accuracy trade-off. We provide more details on those classifiers in Appendix F. We note that our results show performances of specific configurations, but other trade-offs can also be achieved.

We begin by examining how various classifiers perform when estimating the conditional coverage quantity. This is achieved by comparing how well different tabular classifiers estimate the conditional miscoverage. To pursue this, we select the eight largest regression datasets in TabArena (Erickson et al., 2025). Each dataset is divided into three parts; a training set (with $40\%$ of the data) used to learn a

---

[2]covmetrics: `https://github.com/ElSacho/covmetrics` and experiments: `https://github.com/ElSacho/Conditional_Coverage_Estimation`.

predictor $f$ that minimizes the empirical mean squared error. A calibration set (with $10\%$ of the data) used with the non-conformity score $S(X, Y) = |Y - f(X)|$ to construct the set rule $C_\alpha(\cdot)$ with $1 - \alpha = 0.9$ such that when the residual distribution is heteroskedastic, these sets are not expected to be conditional. A test set with $50\%$ of the data, subsampled to different sizes, used to evaluate the $L_1$-ERT, $L_2$-ERT and KL-ERT metrics, performing 5-fold cross-validation.

We compare the estimated metric values as a function of the number of test samples and average all results over ten runs. Since our estimator gives a lower bound on the true $\ell$-ERT value, a larger estimate indicates a stronger classifier. In Table 2 we report the average percentage improvement over the best strategy, averaged across all test sample sizes and all datasets, as well as the average time required to estimate our metrics per 1,000 samples. Because averaging can hide important effects, we also report results for each dataset as a function of the number of test samples in Figures I.3 and I.4.

*Table 2.* **ERT recovered by different methods**, of the maximum recovered ERT among all methods and number of samples, averaged over all number of test samples and datasets. Experiments are repeated 10 times, and the index number is the standard deviation across those 10 experiments. We also report the computational time of the WSC metric.

| Classifier | Avg. % of max ERT | | | Avg. time per 1K samples [s] | Device |
|---|---|---|---|---|---|
| | $L_1$-ERT | $L_2$-ERT | KL-ERT | | |
| TabICLv1.1 | $72.7_{1.6}$ | $54.0_{1.4}$ | $59.2_{1.6}$ | $16.0_{0.0}$ | GPU |
| RealTabPFN-2.5 | $72.1_{1.4}$ | $49.7_{1.1}$ | $55.2_{1.4}$ | $9.2_{0.0}$ | GPU |
| CatBoost | $70.3_{2.1}$ | $49.0_{1.3}$ | $54.1_{1.7}$ | $20.2_{0.5}$ | CPU |
| LightGBM | $68.9_{1.8}$ | $46.4_{1.0}$ | $50.9_{1.4}$ | $2.4_{0.0}$ | CPU |
| RandomForest | $67.3_{2.0}$ | $36.9_{2.4}$ | $35.1_{2.2}$ | $4.4_{0.0}$ | CPU |
| ExtraTrees | $66.5_{1.9}$ | $30.3_{2.4}$ | $27.7_{1.9}$ | $3.4_{0.0}$ | CPU |
| PartitionWise | $33.7_{1.9}$ | $10.5_{0.7}$ | $11.0_{0.8}$ | $0.2_{0.0}$ | CPU |
| WSC | - | - | - | $11.6_{0.1}$ | CPU |

**Results.** As shown in Table 2, the tabular foundation models TabICLv1.1 (Qu et al., 2025) and RealTabPFN-2.5 (Grinsztajn et al., 2025) show excellent performance across different metrics. However, they can be very slow without a GPU, and they are generally only usable up to a certain size of datasets (around 100K samples). Gradient-boosted decision trees like CatBoost (Prokhorenkova et al., 2018) and LightGBM (Ke et al., 2017) are closely behind, but they do facilitate fast training on CPUs and are scalable to large datasets. In particular, LightGBM with the relatively cheap configuration from Holzmüller et al. (2024) excels through its training speed while still recovering large ERT values. Therefore, we suggest LightGBM as the default classifier to use with ERT. Random Forest (Breiman, 2001) and ExtraTrees (Geurts et al., 2006) are also fast but exhibit worse results, particularly for the KL-ERT, which heavily penalizes overconfidence, and the $L_2$-ERT. PartitionWise, the strategy used for CovGap, is fastest but performs much worse than the other classifiers, and often detects almost no

miscoverage (see additional results in Appendix J). Overall, our results suggest that results of tabular classification benchmarks transfer approximately to ERT estimation.

**Differences between metrics.** Table 2 also shows that the $L_1$-ERT is considerably easier to estimate than the $L_2$-ERT and KL-ERT. Indeed, for the $L_1$-ERT, the classifiers only need to predict on the right side of $1 - \alpha$ to be optimal, whereas for the others they need to predict the exact probability. Hence, we recommend using the $L_1$-ERT as the default metric.

**Extension to other modalities.** While our primary analysis focuses on tabular data, recasting conditional coverage estimation as a binary classification problem makes the metric inherently modality-agnostic. Our tabular experiments demonstrate that a classifier's ranking on standard general benchmarks reliably transfers to its ERT estimation performance. Consequently, for non-tabular domains, we recommend deploying a state-of-the-art classifier tailored to that specific modality. Because exhaustive benchmarking across all data types is beyond the scope of this work, practitioners can optimize their classifier choice by referring to established, domain-specific benchmarks, such as ImageNet for vision tasks (Russakovsky et al., 2015), Dwivedi et al. (2023) for graphs, and Wang et al. (2024) for text.

### 4.2. Comparison with existing metrics

We illustrate that existing methods can fail to accurately assess conditional coverage in scenarios where our approach succeeds. To this end, we generate a synthetic dataset following (where $X^1$ is the first component of the vector $X$ and $\sigma(x) = 0.5 + |x| + x^2$).

$$Y \sim \mathcal{N}\big(0, \sigma(X^1)\big), \quad \text{with} \quad X \sim \mathcal{U}([-1, 1]^8),$$

Prediction sets are constructed using two different strategies:

- **Standard CP:** Using the nonconformity score $S(X, Y) = |Y|$ within the standard conformal prediction framework using 3,000 i.i.d. samples.
- **Oracle sets:** Using the ground-truth oracle that provides the true conditional quantiles $\alpha/2$ and $1 - \alpha/2$ of the underlying distribution.

The first strategy produces marginally valid but conditionally invalid prediction sets, while the second produces conditional sets by construction.

The first experiment evaluates how many test points are needed to obtain reliable estimates for commonly used metrics (CovGap, WSC) compared to our proposed metrics. We measure each metric as a function of the number of test points on a log scale, computing $L_1$-ERT and

$L_2$-ERT respectively estimating $\mathbb{E}_X[|p(X) - (1-\alpha)|]$ and $\mathbb{E}_X[(1 - \alpha - p(X))^2]$ using 5-fold cross-validation, and show the results in Figure 1.

**Results.** The results are striking: group-based metrics are extremely unaligned with their theoretical values and require large sample sizes to converge. Even with $5,000$ points, they provide nearly identical diagnostics across these two very different scenarios, and WSC exhibits similar instability. By contrast, our metrics adapt rapidly. In particular, $L_1$-ERT stabilizes very quickly, providing reliable estimates of conditional coverage deviation in the naive scenario. $L_2$-ERT needs more samples to converge to its true value but already diagnoses conditional coverage failure with few samples. This is unsurprising because, as explained earlier, the $L_1$ version only requires that the sign of $h(X) - (1-\alpha)$ matches the sign of $p(X) - (1-\alpha)$, whereas the $L_2$ version instead depends on the closeness of $h(X)$ to the true conditional probability $p(X)$, which typically requires more data.

In the scenario with perfect conditional coverage, all of our proposed metrics converge rapidly to values indicating no failure, while WSC continues to struggle even with $50,000$ samples. These results demonstrate that our methods not only provide more accurate diagnostics but also require far fewer samples to detect conditional coverage deviations reliably.

Figure 3 visualizes the data distribution and the induced prediction sets for both strategies with a test dataset of size $1,500$. Precise metric values are reported in Table 3. Since only the first feature is informative, we plot $(X^1, Y)$ while ignoring the remaining features, and we also show $(X^1, h(X))$ to illustrate our estimator's learned conditional coverage, as well as $(X^1, \text{CovGap}(X))$ to illustrate the difference between a partition-wise estimator and our estimator. As expected, our approach clearly identifies regions of under- and over-coverage in the first scenario, and accurately recovers a near-constant predictor equal to $1 - \alpha$ in the oracle setting.

We evaluate the metrics on a test set of size $1,500$, as reported in Table 3. In the first experiment, where the predictive sets are not conditional, certain metrics fail to detect deviations from conditional coverage. This is the case for SSC, HSIC, and Pearson correlation as they rely solely on prediction set sizes, which are uniform across samples. Similarly, metrics based on feature-space clustering (FSC, CovGap) also fail to detect the coverage violation, due to the difficulty of clustering the feature space. The only baselines that identify a conditional coverage failure in this case are the WSC and EOC metrics.

In contrast, under the second (oracle) strategy, all prediction sets satisfy conditional coverage by construction. The WSC

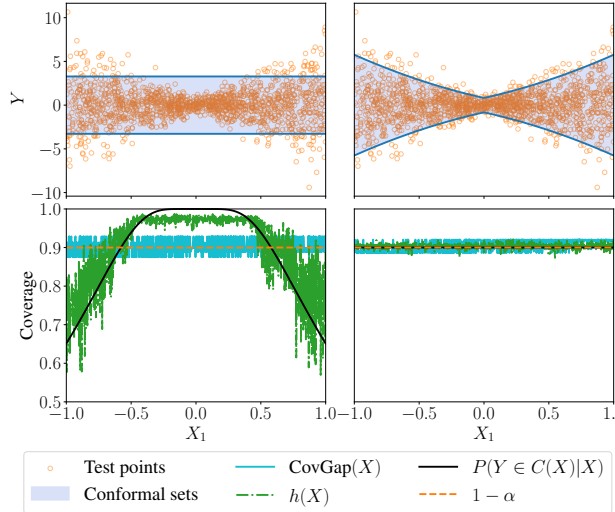

*Figure 3.* Illustration of conditional coverage estimation. The top panel shows data generated as $Y \sim \mathcal{N}(0, \sigma(X^1))$ with $\sigma(x) = 0.5 + |x| + x^2$, where $X \sim \mathcal{U}([-1, 1]^8)$, and $X^1$ is the first component of such a vector $X$. The bottom panel shows the conditional coverage estimation $h$, conditional coverage estimation induced by a partition-wise estimator, true conditional coverage, and desired $1 - \alpha$ conditional coverage. **Left:** Conformal sets from the score $S(X, Y) = |Y|$ using $3,000$ samples. **Right:** Oracle sets that achieve conditional coverage.

metric still reports a conditional coverage failure, with values comparable to the first example. This occurs because, in high-dimensional feature spaces, WSC can overemphasize local fluctuations and misidentify regions with apparent over- or under-coverage. Similarly, EOC detects a false conditional coverage violation by grouping extreme output values together.

Our proposed $L_1$-ERT and $L_2$-ERT metrics, however, correctly distinguish between the two settings: they detect the conditional coverage failure in the first example and report no such failure in the oracle case. Specifically, the $L_1$-ERT of $0.091$ correctly estimates a substantial $9.1\%$ average absolute deviation from the target $90\%$ coverage (closely tracking the true value of $0.098$) and drops close to zero in the conditional case. Furthermore, while the $L_2$-ERT estimate of $0.009$ appears small, this correctly reflects the expected squared deviation, $\mathbb{E}_X[(\mathbb{P}(Y \in C(X)|X) - (1 - \alpha))^2]$. On a root-mean-square scale, this yields $\sqrt{0.009} \approx 0.094$, consistent with the $L_1$ estimate. Crucially, despite this inherent difference in scale, the $0.009$ measured in the non-conditional case remains starkly distinguishable from the $0.000$ achieved under oracle conditional coverage.

### 4.3. Conformal prediction benchmark

We compare conditional coverage metrics across several widely adopted conformal prediction strategies. In particu-

*Table 3.* Conditional metrics for both synthetic samples and whether they accurately diagnose conditional coverage. ✓: Accurate diagnostic. ✗: Failure.
[*] Scale is naturally smaller since $L_2$ measures squared deviations.

|  | Not conditional | Conditional |
|---|---|---|
| **Successfully separates both cases ✓** | | |
| $L_1$-**ERT (Ours)** | $0.091_{0.007}$ ✓ | $-0.005_{0.009}$ ✓ |
| $L_2$-**ERT (Ours)** | $0.009_{0.001}$ ✓[*] | $-0.000_{0.000}$ ✓ |
| **Non-conditional value too close to conditional ✗** | | |
| FSC | $0.868_{0.014}$ ✗ | $0.881_{0.012}$ ✓ |
| CovGap | $0.016_{0.004}$ ✗ | $0.014_{0.004}$ ✓ |
| WCovGap | $0.016_{0.004}$ ✗ | $0.014_{0.004}$ ✓ |
| **Conditional value suggests miscoverage ✗** | | |
| WSC | $0.740_{0.019}$ ✓ | $0.790_{0.014}$ ✗ |
| EOC | $0.341_{0.009}$ ✓ | $0.186_{0.018}$ ✗ |
| **Cannot detect miscoverage for constant set size ✗** | | |
| SSC | $0.004_{0.003}$ ✗ | $0.013_{0.003}$ ✓ |
| HSIC | $0.000_{0.000}$ ✗ | $0.000_{0.000}$ ✓ |
| Pearson | $0.000_{0.000}$ ✗ | $0.019_{0.016}$ ✓ |

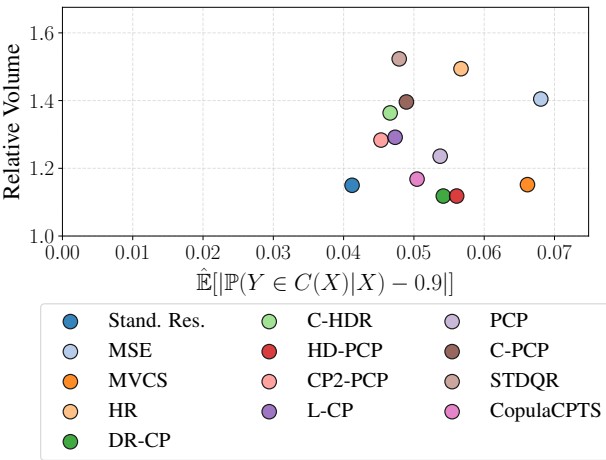

*Figure 4.* Comparison of conditional coverage deviation and normalized prediction set volumes across various conformal prediction strategies ($\alpha = 0.1$). Volumes are scaled by the power $1/d$ and normalized by the minimum observed volume. The $x$-axis displays the $L_1$-ERT metric, which estimates the conditional coverage deviation $\mathbb{E}[|\mathbb{P}(Y \in C(X) \mid X) - (1 - \alpha)|]$.

lar, we build upon the benchmarking framework of Dheur et al. (2025) to evaluate multivariate regression CP methods across 12 datasets (six of which feature multi-dimensional responses).

For each strategy, we partition the data as follows: $60\%$ to train the underlying predictive model, $10\%$ to conformalize the outputs via the standard split CP procedure, and the remaining $30\%$ to evaluate the conditional coverage deviation. This evaluation utilizes various $\ell$-ERT metrics computed via the TabICLv2 classifier (an improved version on the v1.1 used in the previous benchmark) (Qu et al., 2026).[3] Detailed descriptions of all baseline strategies and dataset specifications are provided in Appendices G and H.2, respectively.

All experiments are averaged over ten independent runs. To ensure comparability across diverse target spaces, the prediction region volumes are scaled by the power $1/d$ (where $d$ is the output dimension) and normalized relative to the lowest estimated volume. Figure 4 visualizes these normalized volumes alongside the $L_1$-ERT estimation of the conditional coverage deviation, $\mathbb{E}[|\mathbb{P}(Y \in C(X) \mid X) - (1 - \alpha)|]$. We provide a detailed breakdown of the results for univariate ($Y \in \mathbb{R}$) and multivariate ($Y \in \mathbb{R}^k$, for $k \geq 2$) regression tasks in Appendix I. Overall, our results confirm the expected behavior: methods explicitly designed to target conditional coverage consistently exhibit the lowest conditional deviation.

---
[3] While some CP strategies typically benefit from larger training sets, which could introduce bias if models are under-trained, we allocate a substantial portion of the data ($30\%$) to the test set to ensure a robust and high-fidelity evaluation of the conditional coverage.

## 5. Conclusions

Reliable estimation of conditional coverage is a key challenge for catalyzing further research progress in conformal prediction. We have addressed this challenge by moving from partition-based estimators to a functional-based framework. This shift provides a new perspective on how conditional coverage can be understood, measured, and improved. By framing the problem as one of risk minimization, we introduce a family of interpretable and reliable metrics that leverage the full expressive power of modern predictive models to detect and quantify conditional coverage violations.

Our framework unifies and extends existing diagnostics, transforming what was previously a collection of local, nonparametric estimators into a coherent, model-based approach. This transition not only provides more accurate and stable estimates but also offers a deeper understanding of what conditional coverage represents in practice. While the reliability of our metrics naturally depends on the quality of the learned classifier, this dependency is also a strength.

To encourage reproducibility and practical adoption, we release an open-source package implementing all proposed metrics alongside existing ones. Empirical results confirm the benefits of our approach and reveal that improving conditional coverage often leads to larger prediction sets. Understanding and controlling this balance is a promising direction for future research.

## Acknowledgements

Authors acknowledge funding from the European Union (ERC-2022-SYG-OCEAN-101071601). Views and opinions expressed are however those of the author(s) only and do not necessarily reflect those of the European Union or the European Research Council Executive Agency. Neither the European Union nor the granting authority can be held responsible for them. This publication is part of the Chair "Markets and Learning", supported by Air Liquide, BNP PARIBAS ASSET MANAGEMENT Europe, EDF, Orange and SNCF, sponsors of the Inria Foundation. This work has also received support from the French government, managed by the National Research Agency, under the France 2030 program with the reference "PR[AI]RIE-PSAI" (ANR-23-IACL-0008).

Finally, the authors would like to thank Eugène Berta for fruitful discussions regarding this work.

## Impact Statement

This paper presents work whose goal is to advance the field of Machine Learning. There are many potential societal consequences of our work, none of which we feel must be specifically highlighted here.

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

# Appendices

## Contents

## A. Additional metrics

**Group-based diagnostics.**    We start by reviewing additional group-based diagnostics.

- **Equalized / group-wise coverage.** A fairness-style requirement is that every group attains the nominal coverage:

$$\mathbb{P}_{Y|g(X)=\mathbf{g}}(Y \in C_\alpha(X) \mid g(X) = \mathbf{g}) = 1 - \alpha \qquad \forall\, \mathbf{g} \in \mathcal{G}.$$

  This notion appears in the conformal fairness literature (e.g, Romano et al. 2020a; Ding et al. 2026) and is evaluated in practice by returning the values $C_\mathbf{g}$ for all $\mathbf{g}$.

- **Feature-stratified coverage (FSC).** To focus on the worst-off group, FSC reports the minimal empirical coverage across groups:

$$\mathrm{FSC} \;=\; \min_{\mathbf{g} \in \mathcal{G}} C_\mathbf{g}.$$

  FSC highlights subgroups where coverage is lowest and has been used in several works (e.g, Angelopoulos & Bates 2023; Ding et al. 2023; Jung et al. 2023). This metric is often viewed as a fairness measure, as it focuses on the group that exhibits the poorest coverage.

All of the above group-based diagnostics are sensitive to how the groups $\mathcal{G}$ are chosen. Most of the time, they are created by partitioning the feature space, either with clustering methods, or by using categorical features. Much of the recent work focuses on finding or learning useful partitions that reveal conditional coverage violations, by applying the induced CovGap or FSC metric. In the following, unless otherwise specified, the groups used to evaluate FSC and CovGap are obtained by clustering the feature space. We will use the following notation to refer to alternative grouping strategies.

- **Equal opportunity of coverage (EOC).** To account for differences in outcomes, EOC requires that coverage rates across protected groups are equal conditional on the true label; in regression this means that for each outcome value $y$ (or a discretization of $y$), the coverage within each protected subgroup should match. This idea was introduced by Wang et al. (2023a). However, defining groups based on the outcome can lead to misleading metrics. For instance, consider an interval predictor $[q_{\alpha/2}, q_{1-\alpha/2}]$ for $Y \sim \mathcal{N}(0,1)$, where $q_{\alpha/2}$ and $q_{1-\alpha/2}$ are the $\alpha/2$ and $1-\alpha/2$ quantiles of the normal distribution respectively. If all extreme values of $Y$ are grouped together, the resulting group may show a coverage of zero, even though the prediction interval is conditionally valid.

- **Size-stratified coverage (SSC).** Instead of grouping by features, SSC groups examples by the size (e.g., volume or cardinality) of their prediction sets. It is model-agnostic and useful when $X$ is high-dimensional because it avoids requiring semantically meaningful groups. However, SSC can fail to detect conditional-coverage problems in some settings. Consider, for example regression with the nonconformity score $S(X,Y) := |Y - f(X)|$ (cf. Angelopoulos et al. 2021). This score cannot capture heteroskedasticity (Vovk, 2012), outputting prediction sets with the same sizes independently of the covariate. Thus, in this case, SSC will not reveal coverage failures. Furthermore, it requires access to the prediction set sizes which can be computationally costly for some strategies.

**Representation-based diagnostics (dependence on auxiliary variables).** An alternative to grouping is to measure statistical dependence between the coverage indicator

$$Z := \mathbb{1}\{Y \in C_\alpha(X)\}$$

and auxiliary variables $V$ (for example, prediction-set size, nonconformity score, residuals, or other model-derived quantities). If $Z$ is independent of $V$, this is evidence that coverage does not systematically vary with $V$. Feldman et al. (2021) proposed two measures induced by this remark.

- **Pearson's correlation.** This is a simple measure of linear dependence,

$$R_{\text{corr}}(Z,V) \;=\; \frac{\text{Cov}(Z,V)}{\sqrt{\text{Var}(Z)\,\text{Var}(V)}},$$

where $V$ is the prediction-set size. This is fast and interpretable but only captures linear relationships.

- **HSIC (Hilbert–Schmidt independence criterion).** The HSIC is a nonparametric kernel-based dependence measure that can detect arbitrary nonlinear dependence. Given a suitable pair of kernels, HSIC estimates the maximum mean discrepancy (MMD) (Gretton et al., 2005) between the joint distribution $\mathbb{P}_{Z,V}$ and the product of marginals $\mathbb{P}_Z \otimes \mathbb{P}_V$. Feldman et al. (2021) defined the metric based on HSIC:

$$R_{\text{HSIC}}(Z,V) \;=\; \sqrt{\text{HSIC}(Z,V)},$$

where the square root emphasizes small deviations from independence and gives a loss that is easier to interpret and optimize. Here, $V$ is again the prediction-set size.

**Additional information on the WSC metric** To support our claim that the WSC metric tends to produce pessimistic decisions in the conditional scenario, we performed an experiment isolating its behavior under perfect conditional coverage. The systematic underestimation by the WSC metric stems from an optimization artifact analogous to overfitting. Let $\mathcal{S}$ be the set of considered slabs, and let $C_s$ denote the empirical coverage on a specific slab $s \in \mathcal{S}$. Taking the expectation over draws of the finite dataset, it follows that $\mathbb{E}[\inf_{s' \in \mathcal{S}} C_{s'}] \leq \mathbb{E}[C_s]$ for any fixed $s$. Consequently, $\mathbb{E}[\inf_{s' \in \mathcal{S}} C_{s'}] \leq \inf_{s \in \mathcal{S}} \mathbb{E}[C_s]$. Because the empirical infimum will typically be realized by different slabs depending on the noise in the data draw, this inequality is strictly less. Under true conditional coverage, $\mathbb{E}[C_s] = 1 - \alpha$ for all $s$; therefore, the empirical WSC systematically underestimates the target level $1 - \alpha$.

This artifact becomes particularly pronounced in higher dimensions. It directly reflects the behavior of the theoretical lower bound established in the original WSC formulation (Cauchois et al., 2021), which decreases as a function of the dimension $d$:

$$\inf \text{WSC}_n(\widehat{C}, v) \geq 1 - \alpha - \mathcal{O}(1) \sqrt{\frac{\min\{d, \log |V|\} \log n}{\delta n}}. \tag{4}$$

To empirically corroborate this, we designed a synthetic setup where covariates are drawn as $X \sim \mathcal{U}([0, 1]^d)$, and the coverage indicator $Z \sim \mathcal{B}(1 - \alpha)$ is sampled independently of $X$, ensuring perfect conditional coverage by construction. We evaluated the WSC using $1,000$ slabs across varying dimensions $d$ and sample sizes $n$ (different numbers of slabs yield similar results). As shown in Figure A.1, the average estimated WSC artificially decreases as the input dimension grows. This explicitly confirms that in high-dimensional feature spaces, the WSC metric is structurally pessimistic and prone to reporting spurious conditional coverage violations.

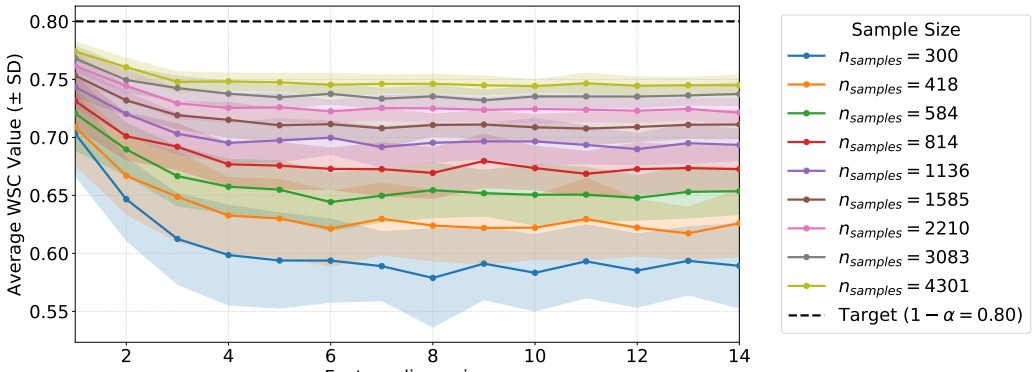

*Figure A.1.* WSC using $1,000$ slabs and $\delta = 0.1$ and miscoverage level $\alpha = 0.2$ across varying dimensions $d$ and sample sizes $n$ for predictive sets achieving conditional coverage.

# B. Background on conformal prediction.

For completeness, we present a brief overview of *split conformal prediction*, a widely used and computationally simple approach for constructing prediction sets with marginal validity guarantees (Papadopoulos et al., 2002; Lei et al., 2018; Angelopoulos & Bates, 2023). Suppose we observe data $\mathcal{D} = \{(X_i, Y_i)\}_{i=1}^n$ sampled i.i.d. from a joint distribution $\mathbb{P}_{X,Y}$, where $X_i \in \mathcal{X}$ are feature vectors and $Y_i \in \mathcal{Y}$ are outcomes. For a new test pair, $(X_{\text{test}}, Y_{\text{test}})$, the aim is to form a predictive set $C_\alpha(X_{\text{test}})$ such that

$$\mathbb{P}_{X,Y}(Y_{\text{test}} \in C_\alpha(X_{\text{test}})) = 1 - \alpha.$$

To achieve this, the dataset is randomly divided into two parts: a *training set* $\mathcal{D}_1$ of size $n_1$ and a *calibration set* $\mathcal{D}_2$ of size $n_2$. A predictive model is fit on $\mathcal{D}_1$, while $\mathcal{D}_2$ is reserved for calibrating the prediction sets. Central to the procedure is a nonconformity score $S(X, Y) \in \mathbb{R}$, which quantifies how atypical a candidate response $Y$ is relative to the model. Using these scores, the predictive sets are then computed as:

$$C_\alpha(X_{\text{test}}) = \{y : S(X_{\text{test}}, y) \leq \hat{q}_\alpha\}, \tag{5}$$

with

$$\hat{q}_\alpha := Q_{1-\alpha} \left( \frac{1}{n_2 + 1} \sum_{k=1}^{n_2} \delta_{S(X_k, Y_k)} + \frac{\delta_\infty}{n_2 + 1} \right),$$

where $Q_{1-\alpha}(\mathbb{P})$ returns the $1 - \alpha$ quantile of the distribution $\mathbb{P}$, and $\delta_x$ is the Dirac measure centered at $x$. By the exchangeability of the data, this construction guarantees the marginal coverage property:

$$\mathbb{P}_{X,Y}(Y_{\text{test}} \in C_\alpha(X_{\text{test}}) \mid \mathcal{D}_1) \in \left[1 - \alpha, 1 - \alpha + \frac{1}{n_2 + 1}\right).$$

It is important to note that the guarantee in Eq. (5) is marginal, which means that coverage holds on average over the distribution of $X_{\text{test}}$ and $\mathcal{D}_2$. A stronger requirement is *conditional coverage*, which demands

$$\mathbb{P}_{Y|X}(Y_{\text{test}} \in C_\alpha(X_{\text{test}}) \mid X_{\text{test}}) = 1 - \alpha, \tag{6}$$

for almost every $X_{\text{test}}$, but achieving (6) is impossible in general without additional assumptions. For a given conformal prediction strategy that achieves marginal coverage (5), it is essential to be able to measure how close from a conditional guarantee (6) we are.

## C. Estimating Bregman divergences with ERTs

The following proposition is related to Proposition 3.1 and studies the setting when there is a single proper score $\ell$ that estimates a distance $d(1 - \alpha, p)$ simultaneously for all $\alpha$ and $p$.

**Proposition C.1.** *A function $d(p, q)$ arises as the divergence of some proper scoring rule if and only if there exists a convex function $\varphi : [0, 1] \to \mathbb{R}$ such that*

$$d(p, q) = D_\varphi(q\|p) = \varphi(q) - \varphi(p) - (q - p)\, s(p)$$

*for any choice of subgradient $s(p) \in \partial\varphi(p)$. Furthermore, the associated proper score is defined as*

$$\ell_\varphi(p, y) := -\varphi(p) - (y - p)\, s(p).$$

*and*

$$\ell_\varphi - \text{ERT} = \mathbb{E}_X[D_\varphi(p(X)\|1 - \alpha)].$$

*Proof.* Let $\ell$ be a proper scoring rule for binary outcomes and write $p, q \in [0, 1]$ for probabilities of the event $Y = 1$. Define

$$\varphi(q) := -\mathbb{E}_{Y \sim q}\big[\ell(q, Y)\big] = -q\, \ell(q, 1) - (1 - q)\, \ell(q, 0).$$

Properness of $\ell$ means that for every fixed $p \in [0, 1]$ and every $q \in [0, 1]$,

$$\varphi(q) = -\mathbb{E}_{Y \sim q}[\ell(q, Y)] \geq -\mathbb{E}_{Y \sim q}[\ell(p, Y)] = -q\, \ell(p, 1) - (1 - q)\, \ell(p, 0).$$

Rearranging gives

$$\varphi(q) \geq \varphi(p) + (q - p)(\ell(p, 0) - \ell(p, 1))$$

so $\varphi$ is convex on $[0, 1]$ and the function $p \mapsto \ell(p, 0) - \ell(p, 1)$ defines a subgradient of $\varphi$ at $p$. Denote by $s(p) \in \partial\varphi(p)$ any such subgradient. Then for every $p, q \in [0, 1]$,

$$\mathbb{E}_q[\ell(p, Y)] = -\varphi(p) - (q - p)\, s(p),$$

and hence the divergence of $\ell$ satisfies

$$d_\ell(p, q) = \mathbb{E}_q[\ell(p, Y)] - \mathbb{E}_q[\ell(q, Y)] = -\varphi(p) - (q - p)\, s(p) + \varphi(q) = \varphi(q) - \varphi(p) - (q - p)\, s(p),$$

which is exactly the Bregman divergence $D_\varphi(q\|p)$ generated by $\varphi$.

Conversely, let $\varphi : [0, 1] \to \mathbb{R}$ be convex and for each $p \in [0, 1]$ choose a subgradient $s(p) \in \partial\varphi(p)$. Define a score $\ell_\varphi$ by

$$\ell_\varphi(p, y) := -\varphi(p) - (y - p)\, s(p).$$

by construction, and convexity of $\varphi$ implies for every $p$ and $q$,

$$\mathbb{E}_q[\ell_\varphi(p, Y)] = -\varphi(p) - (q - p)\, s(p) \geq -\varphi(q) = \mathbb{E}_q[\ell_\varphi(q, Y)].$$

Thus $\ell_\varphi$ is proper, and its divergence equals

$$\mathbb{E}_q[\ell_\varphi(p, Y)] - \mathbb{E}_q[\ell_\varphi(q, Y)] = -\varphi(p) - (q - p)\, s(p) + \varphi(q) = D_\varphi(q\|p).$$

Therefore a function $d(p, q)$ arises as the divergence of a proper scoring rule if and only if it is the Bregman divergence of some convex generator $\varphi$, as claimed. Note that the score $\ell_\varphi$ is determined up to addition of an arbitrary function of the outcome whose expectation under every $q$ is zero, and that when $\varphi$ is differentiable the subgradient $s(p)$ may be replaced by the derivative $\varphi'(p)$.

Using this proper score to evaluate the $\ell_\varphi$-ERT we get (with Z the binary outcome as defined in the main text)

$$
\begin{aligned}
\ell_\varphi\text{-ERT} &= \mathcal{R}_{\ell_\varphi}(1 - \alpha) - \mathcal{R}_{\ell_\varphi}(p) \\
&= \mathbb{E}_{X,Z}[\ell_\varphi(1 - \alpha, Z) - \ell_\varphi(p(X), Z)] \\
&= \mathbb{E}_X[\mathbb{E}_Z[\ell_\varphi(1 - \alpha, Z) - \ell_\varphi(p(X), Z)|X]] \\
&= \mathbb{E}_X[\mathbb{E}_{Z\sim p(X)}[\ell_\varphi(1 - \alpha, Z) - \ell_\varphi(p(X), Z)]] \\
&= \mathbb{E}_X[d_{\ell_\varphi}(1 - \alpha, p(X))] \\
&= \mathbb{E}_X[D_\varphi(p(X)\|1 - \alpha)]. \qquad\qquad \square
\end{aligned}
$$

**Proof of Proposition 3.1.**

*Proof.* First, $\ell$ is a proper score because for all $p, q \in [0, 1]$, convexity of $f$ yields

$$
\mathbb{E}_{y\sim q}[\ell(p, y)] = -f(p) - (q - p)f'(p) \geq -f(q) = \mathbb{E}_{y\sim q}[\ell(q, y)] .
$$

Since we assumed $f'(1 - \alpha) = 0$, we have $\ell(1 - \alpha, Z) = -f(1 - \alpha) = 0$. Hence,

$$
\mathbb{E}_{Z\sim p}[\ell(1 - \alpha, Z) - \ell(p, Z)] = \mathbb{E}_{Z\sim p}[-\ell(p, Z)] = f(p) + (p - p)f'(p) = f(p) .
$$

Taking the expectation over $X$ yields the claim. $\qquad\qquad \square$

## D. Link with Gibbs et al. (2025)

To get predictive sets with conditional guarantees, Gibbs et al. (2025) used the fact that conditional coverage holds if for all measurable function $\varphi$,

$$
\mathbb{E}_{X,Z}[\varphi(X)(1 - \alpha - Z)] = 0.
$$

To obtain a more operational interpretation, suppose we restrict attention to predictors of the form $h(x) = \theta^T \phi(x)$, where $\phi(x)$ is a feature map. The associated mean squared risk is

$$
F(\theta) = \mathbb{E}_{X,Z}[(\theta^T \phi(X) - Z)^2]
$$

and has gradient

$$
\nabla_\theta F(\theta) = 2\,\mathbb{E}_{X,Z}[(\theta^T \phi(X) - Z)\phi(X)].
$$

Write $\theta_\alpha$ such that $h_{\theta_\alpha}(X) = 1 - \alpha$, (that exists when $\phi(X)$ has a non-zero constant component).

For this class of models, our metric $L_2$-ERT compares any $\theta$ to this target via

$$
F(\theta_\alpha) - \inf_\theta F(\theta).
$$

When conditional coverage holds, both the gradient and the $L_2$-ERT are equal to zero. In this sense, the quantity $F(\theta_\alpha) - F(\theta)$ offers more interpretability than the gradient alone. While the gradient describes only a local direction in the parameter space of improvement, the risk difference has a clear interpretation that has already been discussed. Indeed, when $F(\theta_\alpha) - \inf_\theta F(\theta) = 0$ we have $\mathbb{E}_{X,Z}[(1 - \alpha - Z)\phi(X)] = 0$ but it can be that $F(\theta_\alpha) - \inf_\theta F(\theta)$ is very small but that $\|\mathbb{E}_{X,Z}[(1 - \alpha - Z)\phi(X)]\|_2$ is very large. That is why $\mathbb{E}_{X,Z}[\varphi(X)(1 - \alpha - Z)]$ cannot be used as an interpretable quantity to quantify conditional miscoverage deviation, but only to assess if there is conditional coverage or not.

# E. Extension to a conditional coverage rule

While conditional coverage is often defined with a fixed target level,

$$\mathbb{P}\big(Y \in C_\alpha(X_{\text{test}}) \mid X_{\text{test}}\big) = 1 - \alpha \quad \mathbb{P}_X\text{-almost surely,}$$

ensuring that the prediction set $C_\alpha(X)$ covers the true label with probability $1 - \alpha$ for every possible input $X$, a uniform coverage level may be neither necessary nor desirable in practice. In many settings, one may wish to adapt the coverage level to reflect varying uncertainty, heteroskedastic noise, or task-specific risk preferences. Recent conformal prediction methods allow adapting $\alpha$ dynamically to optimize other objectives (Gauthier et al., 2025; 2026).

To formalize this flexibility, we introduce a *conditional miscoverage rule* $\alpha : \mathcal{X} \to [0, 1]$, which prescribes a desired miscoverage level $\alpha(X)$ that may vary with the input features. Under this generalized framework, the target conditional coverage condition becomes

$$\mathbb{E}\big[\mathbb{1}\{Y \in C_\alpha(X_{\text{test}})\} \mid X_{\text{test}}\big] = 1 - \alpha(X_{\text{test}}) \quad \mathbb{P}_X\text{-almost surely.}$$

This formulation recovers the standard conformal setting in the special case where $\alpha(X)$ is constant, while enabling the analysis of predictors designed to achieve non-uniform, data-dependent coverage guarantees. It thus provides a principled way to study how prediction methods align with arbitrary, application-driven notions of conditional reliability. This could be useful for example in classification, where the discrete nature of the target may not allow to achieve constant coverage.

We next adapt our framework for the $\ell$-ERT metric under this setting by writing

$$\ell\text{-ERT} = \mathcal{R}_\ell(1 - \alpha) - \mathcal{R}_\ell(p) = \mathbb{E}\Big[\big(d_\ell(1 - \alpha(X), p(X))\big)\Big].$$

and its variational form

$$\ell\text{-ERT}(h) = \mathcal{R}_\ell(1 - \alpha) - \mathcal{R}_\ell(h) = \mathbb{E}_{X,Y}\Big[\ell(1 - \alpha(X), Y) - \ell(h(X), Y)\Big].$$

The formulas above work for the $L_2$-ERT and the KL-ERT, where the proper score $\ell$ does not depend on $1 - \alpha$. If general distance metrics should be estimated as in Proposition 3.1, such as for the $L_1$-ERT, the proper scoring rule $\ell$ needs to depend on $\alpha(X)$, and we obtain

$$\ell\text{-ERT} = \mathbb{E}\Big[\big(d_{\ell_{\alpha(X)}}(1 - \alpha(X), p(X))\big)\Big].$$

and its variational form

$$\ell\text{-ERT}(h) = \mathbb{E}_{X,Y}\Big[\ell_{\alpha(X)}(1 - \alpha(X), Y) - \ell_{\alpha(X)}(h(X), Y)\Big].$$

The remaining components of the procedure remain unchanged. For convenience, we refer to the resulting metrics using the same terminology as in the fixed $1 - \alpha$ case, since that setting corresponds to a particular instance of this more general framework.

# F. Strong classifiers

Here, we provide more details on the classifiers used in Section 4. In the following, we will refer to training and test data for the data that the classifier $h$ is trained and evaluated on, not to be confused with the data that the original prediction set method $C_\alpha$ is trained on. We employ the following classifiers in our evaluation:

**Tabular foundation models.** We evaluate the recent models **RealTabPFN-2.5** (Grinsztajn et al., 2025) and **TabICLv1.1** (Qu et al., 2025). These models can predict $h(x_1^{\text{test}}), \ldots, h(x_n^{\text{test}})$ with a single forward pass through a neural network that takes both the test input and the entire training set into account. They have been found to perform very well already without hyperparameter tuning (Erickson et al., 2025).

**Gradient-boosted decision trees.** We choose two representatives: CatBoost (Prokhorenkova et al., 2018) is known for its strong default performance. We adopt its hyperparameters from AutoGluon (Erickson et al., 2020) and TabArena (Erickson et al., 2025), using 300 early stopping rounds instead of AutoGluon's custom early stopping logic. To obtain a faster model, we use LightGBM (Ke et al., 2017) with cheaper hyperparameters adapted from the tuned defaults of Holzmüller et al. (2024), reducing the number of early stopping rounds to 100 and using cross-entropy loss for early stopping (as for CatBoost). Both CatBoost and LightGBM are fitted in parallel for eight inner cross-validation folds for each outer cross-validation fold, following TabArena. The inner cross-validation folds are used for early stopping, and the final validation predictions are concatenated and used to fit a quadratic scaling post-hoc calibrator (Berta et al., 2026). Test set predictions are made based on the post-hoc calibrator applied to the average of the eight models' predictions.

**Bagging models.** Random Forest (Breiman, 2001) is a popular baseline for tabular ML. Extremely randomized trees (ExtraTrees/XT, Geurts et al., 2006) are a more randomized variant that performs similarly while being faster (Erickson et al., 2025). We fit both models with 300 estimators and otherwise use the default hyperparameters from `scikit-learn` (Pedregosa et al., 2011).

**PartitionWise** Partition-wise estimation first groups samples in the feature space and then predicts by using the average label within each group. The predictor relies on KMeans to form the partitions, and the number of clusters is selected through a fourth root rule based on the test set size, which keeps the model flexible while avoiding clusters that are too small. At inference time, each new sample is assigned to its nearest cluster and the model predicts the mean stored for that cluster.

**Trade-offs.** We chose to only fit boosted trees with inner cross-validation, both because they need validation sets for early stopping and because they are still reasonably fast with parallelization. The use of cross-validation also allows for the application of post-hoc calibration. Other methods might also benefit from post-hoc calibration, especially for non-L1 metrics, at the cost of higher runtime due to cross-validation.

**Discussion and other options.** We omit from-scratch trained tabular neural networks from our comparison as they are relatively slow, especially on CPUs, and their un-tuned performance is suboptimal (Erickson et al., 2025). If runtime is less of a concern, for ideal sample-efficiency, automated machine learning methods such as AutoGluon (Erickson et al., 2020) can be employed that combine multiple models and hyperparameter setting, ensembling, and post-hoc calibration. However, when these methods are used with time limits, the result may not be reproducible.

**Hardware.** We ran tabular foundation models on GPUs (NVIDIA V100) and the other models on CPUs (Cascade Lake Intel Xeon 5217 with 8 cores).

# G. Benchmarking strategies

The strategies we compare can be differentiated in four main groups—those that uses density estimation, latent spaces, hyper-rectangles, or minimizing a quantity while ensuring marginal coverage. We build upon the work of Dheur et al. (2025) that already explained most of those strategies, but we recall their specificities here for completeness.

Among the density-based methods, given a predictive density $\hat{p}(y|x)$, the benchmarked strategies are:

- **DR-CP** (Sadinle et al., 2019): Defines the conformity score as $S_{\text{DR-CP}}(X, Y) = -\hat{p}(Y|X)$, leading to prediction regions that are density superlevel sets, $C_{\text{DR-CP}}(X) = \{y : \hat{p}(y|X) \geq -\hat{q}\}$.

- **C-HDR** (Izbicki et al., 2022): Conformalize the highest predictive density (HPD) by using the nonconformity score $S_{\text{HDP}}(X, Y) = \mathbb{P}_{y \sim \hat{p}(\cdot|X)}(\hat{p}(y|X) \geq \hat{p}(Y|X))$. It then produces regions $C_{\text{C-HDR}}(X) = \{y : \hat{p}(y|X) \geq \hat{t}_q\}$, where $\hat{t}_q$ defines the highest density region (HDR) at level $\hat{q}$.

- **PCP** (Wang et al., 2023b): Draws $L$ samples $\tilde{Y}^{(l)} \sim \hat{p}_{Y|x}$ with $\hat{p}_{Y|x}$ the estimated conditional distribution, and defines conformity as the distance to the nearest sample, $S_{\text{PCP}}(X, Y) = \min_{l \in [L]} \|Y - \tilde{Y}^{(l)}\|_2$; the corresponding region is a union of $L$ balls centered at the sampled points.

- **HD-PCP** (Wang et al., 2023b): Extends PCP by retaining only the top $\lfloor(1-\alpha)L\rfloor$ samples with highest density, concentrating the prediction region on high-density areas.

- **C-PCP** (Dheur et al., 2025): Estimates the conditional CDF of the conformity score $S(X, Y)$,

$$S_{\text{CDF}}(x, y) = \mathbb{P}(S_W(X, Y) \leq S_W(x, y) \mid X = x),$$

using a Monte Carlo approximation with $K$ samples

$$S_{\text{ECDF}}(x, y) = \frac{1}{K} \sum_{k=1}^{K} \mathbb{1}[S_W(x, \hat{Y}^{(k)}) \leq S_W(x, y)], \quad \hat{Y}^{(k)} \sim \hat{F}_{Y|x}.$$

When $S(x, y) = S_{\text{PCP}}(x, y)$, this yields

$$S_{\text{C-PCP}}(x, y) = \frac{1}{K} \sum_{k \in [K]} \mathbb{1}\left\{ \min_{l \in [L]} \|\hat{Y}^{(k)} - \tilde{Y}^{(l)}\|_2 \leq \min_{l \in [L]} \|y - \tilde{Y}^{(l)}\|_2 \right\}.$$

- **CP2-PCP** (Plassier et al., 2025b): Builds predictive sets by using samples from an implicit conditional generative model. For each calibration point it uses two independent draws from the conditional generator to define a conformity score and an inflation parameter $\tau$ that accounts for the conditional mass around likely outputs. At prediction time it forms a union of balls around new generated samples, with their size chosen to guarantee marginal validity while improving approximate conditional adaptivity.

- **Standardized residuals** (Lei et al., 2018; Braun et al., 2025b): Normalizes the residuals by their estimated conditional covariance matrix $\Sigma(X)$ with the score

$$S_{\text{Stand. Res.}}(X, Y) = \|\Sigma(X)^{-1/2}(Y - f(X))\|_2.$$

- **MSE**: Naïve multivariate generalization of the *univariate* score $S(X, Y) = |Y - f(X)|$, by using the *multivariate* score $S(X, Y) = \|Y - f(X)\|_2$.

Among the latent space-based methods, the benchmarked strategies are:

- **STDQR** (Feldman et al., 2023): Constructs multivariate prediction regions in a latent space $\mathcal{Z}$ to overcome limitations of standard multivariate prediction methods. Instead of using directional quantile regression as originally introduced, we follow Dheur et al. (2025) procedure where the region $R_{\mathcal{Z}}$ with coverage $1 - \alpha$ is constructed by selecting the $1 - \alpha$ proportion of latent samples closest to the origin, ensuring correct coverage directly in the latent space. These latent regions are then mapped to the output space $\mathcal{Y}$ via a conditional generative model (originally a CVAE, here replaced with a normalizing flow). A conformalization step refines coverage by creating a grid of latent samples, mapping them to $\mathcal{Y}$, and forming small balls around each mapped point.

- **L-CP** (Dheur et al., 2025): Defines conformity in a latent space using an invertible conditional generative model $\hat{Q} : \mathcal{Z} \times \mathcal{X} \to \mathcal{Y}$. A latent variable $Z \sim \mathcal{N}(0, I_d)$ is mapped to the output space via $\hat{Q}$, and the conformity score is measured in latent space as

$$S_{\text{L-CP}}(X, Y) = \|\hat{Q}^{-1}(Y; X)\|.$$

The prediction region is obtained by taking a ball of radius $\hat{q}$ around the origin in latent space and mapping it back to the output space. This method avoids grid-based directional quantile regression, improving scalability and computational efficiency, and generalizes distributional conformal prediction to multivariate outputs.

Among the hyper-rectangle-based methods, the benchmarked strategies are:

- **CopulaCPTS** (Sun & Yu, 2024): This method models the joint dependence between marginal conformity scores via a copula. The calibration data are split into two sets: $\mathcal{D}_{\text{cal-1}}$ to estimate empirical CDFs $\hat{F}_i$ of conformity scores for each output dimension $i \in [d]$, and $\mathcal{D}_{\text{cal-2}}$ to calibrate the copula parameters. The optimal thresholds $s_1^*, \ldots, s_d^*$ are obtained by minimizing a coverage-based loss, ensuring marginal validity while reducing region size. The final prediction region is

$$C_{\text{CopulaCPTS}}(X) = \{y \in \mathcal{Y} : S_i(X, y_i) < s_i^*, \ \forall i \in [d]\}.$$

Other copula's based strategies include Messoudi et al. (2021); Mukama et al. (2025).

- **HR** (Romano et al., 2019; Zhou et al., 2024): Constructs axis-aligned (hyper-rectangular) prediction regions by fitting univariate quantiles $\tilde{q}_{\alpha/2}(x)_i$ and $\tilde{q}_{1-\alpha/2}(x)_i$ following Romano et al. (2019) for each output dimension $i \in [k]$, with $\tilde{\alpha} = 1 - (1-\alpha)^{1/k}$. The conformity score is defined as (inspired from Zhou et al. (2024) which extends uni-variate scorings to multi-variate ones)

$$S_{\text{HR}}(X, Y) = \max_{i \in [k]} \left\{ \tilde{q}_{\alpha/2}(X)_i - Y_i, \ Y_i - \tilde{q}_{1-\alpha/2}(X)_i \right\},$$

yielding rectangular prediction regions aligned with coordinate axes.

Finally, among the strategies which minimizes the size of the prediction sets, we use:

- **MVCS** (Braun et al., 2025a): Minimizes the volume of the sets $\{y \in \mathbb{R}^k, \|M(X)(y - f(X))\|_p \leq 1\}$ where $M(X)$ is positive definite, $f(X) \in \mathbb{R}^k$, and $p > 0$ defines a $p$-norm, while ensuring valid marginal coverage. The conformalization set is done with the score $S(X, Y) = \|M(X)(Y - f(X))\|_p$.

## H. Additional information regarding the experiments

### H.1. Details on WSC

For our experiments, with fixed $\delta = 0.1$.

### H.2. Details on the datasets

See Table H.1 for details on the datasets used for the classifiers comparison, Table H.2 for details on the uni-variate datasets and Table H.3 for the multivariate ones.

*Table H.1.* Description of the univariate datasets.

| Dataset | Number of samples | Number of test samples | Number of features |
|---|---|---|---|
| physiochemical_protein | 45730 | 22865 | 9 |
| Food_Delivery_Time | 45593 | 22797 | 10 |
| diamonds | 53940 | 26970 | 9 |
| superconductivity | 21263 | 10632 | 81 |
| ailerons | 13750 | 6875 | 40 |
| o11 | 5742 | 2872 | 1025 |
| miami2016 | 13932 | 6967 | 16 |
| winequality | 6497 | 3250 | 12 |

*Table H.2.* Description of the univariate datasets.

| Dataset | Number of samples | Number of test samples | Number of features |
|---|---|---|---|
| ailerons | 13750 | 4125 | 40 |
| bank8FM | 8192 | 2458 | 8 |
| cpu-act | 8192 | 2458 | 21 |
| house-8L | 22784 | 6836 | 8 |
| miami | 13932 | 4182 | 16 |
| sulfur | 10081 | 3027 | 6 |

*Table H.3.* Description of the multivariate datasets.

| Dataset | Number of samples | Number of test samples | Number of features | Dimension of targets |
|---------|-------------------|------------------------|--------------------|----------------------|
| Bias | 7752 | 2326 | 22 | 2 |
| CASP | 45730 | 13719 | 8 | 2 |
| House | 21613 | 6484 | 17 | 2 |
| rf1 | 9125 | 2738 | 64 | 8 |
| rf2 | 9125 | 2738 | 576 | 8 |
| Taxi | 61286 | 18386 | 6 | 2 |

# I. Additional experiments

**Over- and under-coverage**    To illustrate our findings regarding over- and under-coverage of Section 3.3, we performed one experiment in the synthetic data regime. To do so, we reuse the experimental setup that has been used in Section 4.2, that are the standard CP setup and the oracle one. For the oracle setup, the sets therefore provides a conditional coverage of $0.90$. We choose to compare this conditional coverage with the target level $0.80$. The sets are therefore all over-covering, by $0.10$, and we expect our metric for the over-coverage $L_1^+$-ERT to be equal to $0.10$ (as $0.90 - 0.80 = 0.10$), but the under-coverage $L_1^-$-ERT to be equal to $0$, as the sets are never under-covering. This is indeed what we observe empirically in Figure I.1.

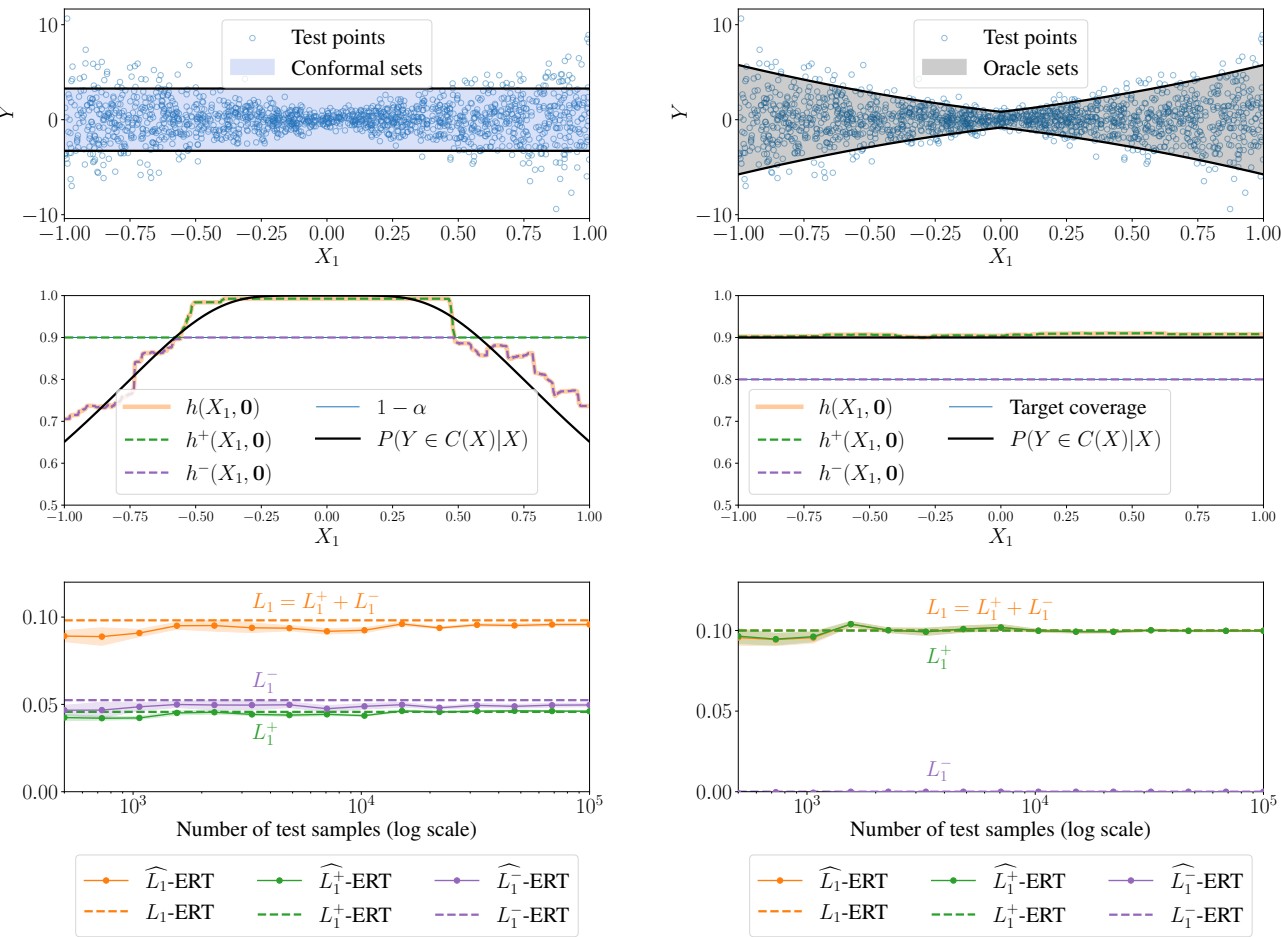

*Figure I.1.* Illustration of conditional over- and under-coverage estimation. The top panel shows data generated as $Y \sim \mathcal{N}(0, \sigma(X^1))$ with $\sigma(x) = 0.5 + |x| + x^2$, where $X \sim \mathcal{U}([-1, 1]^8)$, and $X^1$ is the first component of such a vector $X$. The middle panel shows the conditional coverage estimation $h$ (with the non-informative features fixed to zero for clarity), and its truncated version to $1 - \alpha$, $h^+$ and $h^-$. We also plot the true conditional coverage, and the target coverage. All estimations are performed using $1,500$ test points. The bottom panel shows the estimated $L_1$-ERT, and its decomposition between $L_1^+$-ERT for over-coverage and $L_1^-$-ERT for under-coverage. We do observe that $L_1 = L_1^+ + L_1^-$. **Left:** Conformal sets from the score $S(X, Y) = |Y|$ using $3,000$ samples. **Right:** Oracle sets that achieve conditional coverage of $0.90$, but tested with a target coverage of $0.80$.

**Robustness of the metric.**    To explicitly isolate and evaluate the robustness of our conditional coverage estimation, we design two synthetic stress-test scenarios. Our inferential goal is to test the estimator's capacity to capture both smooth and high-frequency heteroscedasticity profiles.

In both experiments, we sample 8-dimensional covariates $X \sim \mathcal{U}([-1, 1]^8)$ and generate response variables $Y \sim \mathcal{N}(0, \sigma(X^1))$. The data-generating process relies exclusively on the first feature, $X^1$, embedding the true conditional signal within seven non-informative dimensions. We construct baseline prediction sets using a naive non-conformity score, $S(X, Y) = |Y|$, calibrated on $3,000$ samples. Because this score ignores local dispersion, the resulting regions achieve

marginal validity but exhibit strong conditional coverage violations. We then compare our $L_1$-ERT metric with a LightGBM classifier against a standard partition-wise estimator (analogous to CovGap).

We evaluate robustness across two distinct variance profiles in Figure I.2:

- **Robustness to smooth variations and noise (left):** We define a V-shaped variance profile, $\sigma(x) = 0.5 + 0.25|x|$, and evaluate on $1,500$ test points. The explicit presence of non-informative features structurally degrades partition-based methods, which suffer from volume explosion and binning inefficiencies in $\mathbb{R}^8$. Empirically, we observe that the LightGBM classifier successfully isolates the informative feature $X^1$, accurately reconstructing the smooth conditional coverage deficit $h(X)$, whereas the partition-wise approach degrades.

- **Robustness to high-frequency variations (right):** We impose a highly non-smooth, localized variance profile: $\sigma(x) = 0.5 + \mathbf{1}\{x \in \cup_{0 \le k \le 9}[\frac{k-5}{5}, \frac{k-5}{5} + 0.04]\}$, evaluated over $3,000$ test points. This setup tests the spatial resolution limits of the estimators. Partition-wise estimators structurally fail to capture these sharp, rapid oscillations due to rigid binning artifacts. Conversely, the LightGBM demonstrates a superior capacity to track localized drops in coverage without overfitting to the high-dimensional noise.

Consequently, the empirical evidence supports the conclusion that, with the right classifier, the $L_1$-ERT provides a robust evaluation of conditional coverage deviation in the presence of complex heteroscedasticity and high-dimensional nuisance parameters.

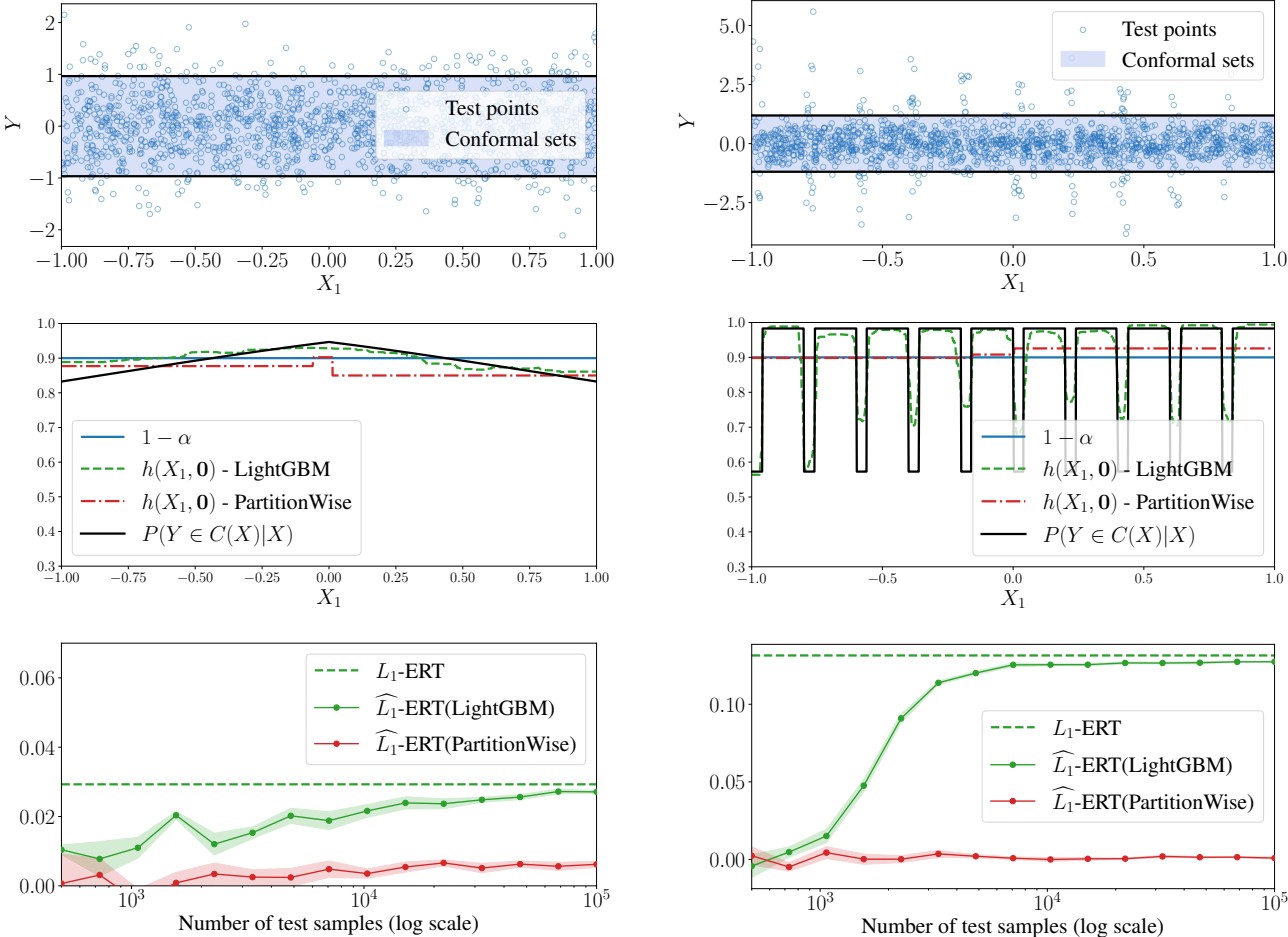

*Figure I.2.* **Left:** Illustration of conditional coverage estimation. The top panel shows data generated as $Y \sim \mathcal{N}(0, \sigma(X^1))$ with $\sigma(x) = 0.5 + 0.25|x|$, where $X \sim \mathcal{U}([-1, 1]^8)$, and $X^1$ is the first component of such a vector $X$. Sets are obtained with conformal prediction from the score $S(X, Y) = |Y|$ using $3,000$ samples. The middle panel shows the conditional coverage estimation $h$ (with the non-informative features fixed to zero for clarity) with a LightGBM classifier and a partition-wise estimator, to mimic the estimation provided by a CovGap estimate. All estimations are performed using $1,500$ test points. We also plot the true conditional coverage, and the target coverage. The bottom panel shows the estimated $L_1$-ERT estimated with a LightGBM or a partition-wise estimator. **Right:** Illustration of conditional coverage estimation. The top panel shows data generated as $Y \sim \mathcal{N}(0, \sigma(X^1))$ with $\sigma(x) = 0.5 + \mathbf{1}\{x \in \cup_{0 \le k \le 9}[\frac{k-5}{5}, \frac{k-5}{5} + 0.04]\}$, where $X \sim \mathcal{U}([-1, 1]^8)$, and $X^1$ is the first component of such a vector $X$. Sets are obtained with conformal prediction from the score $S(X, Y) = |Y|$ using $3,000$ samples. The middle panel shows the conditional coverage estimation $h$ (with the non-informative features fixed to zero for clarity) with a LightGBM classifier and a partition-wise estimator, to mimic the estimation provided by a CovGap estimate. All estimations are performed using $3,000$ test points. We also plot the true conditional coverage, and the target coverage. The bottom panel shows the estimated $L_1$-ERT estimated with a LightGBM or a partition-wise estimator.

**Tabular benchmark.** We present the disaggregated results of the tabular classifier benchmark across individual datasets. To maintain epistemic clarity, we explicitly decompose the computation of the "Avg. % of max ERT" metric reported in Table 2. Our inferential goal with this metric is to establish a scale-invariant evaluation framework. Because raw ERT magnitudes can vary drastically depending on the dataset's underlying noise and dimensionality, directly averaging raw scores would implicitly over-weight datasets with inherently larger baseline errors.

To correct for this confounding factor, the metric is computed via the following explicit sequence:

1. For each distinct experiment and dataset, we isolate the empirical supremum (maximum estimated ERT) across all evaluated tabular models and test set sizes.

2. We map the raw ERT of each specific model and test-size configuration to a normalized percentage relative to this local maximum.

3. Finally, we marginalize over the task dimensions by computing the arithmetic mean of these normalized percentages across all experiments and datasets.

We present the estimated $\ell$-ERT for various classifiers on eight datasets, plotted against the number of test samples in Figures I.3 & I.4 & I.5 & I.6 & I.7 & I.8 & I.9 & I.10 & I.11 & I.12 & I.13 & I.14. The plots highlight a pronounced difference between PartitionWise and the remaining classifiers.

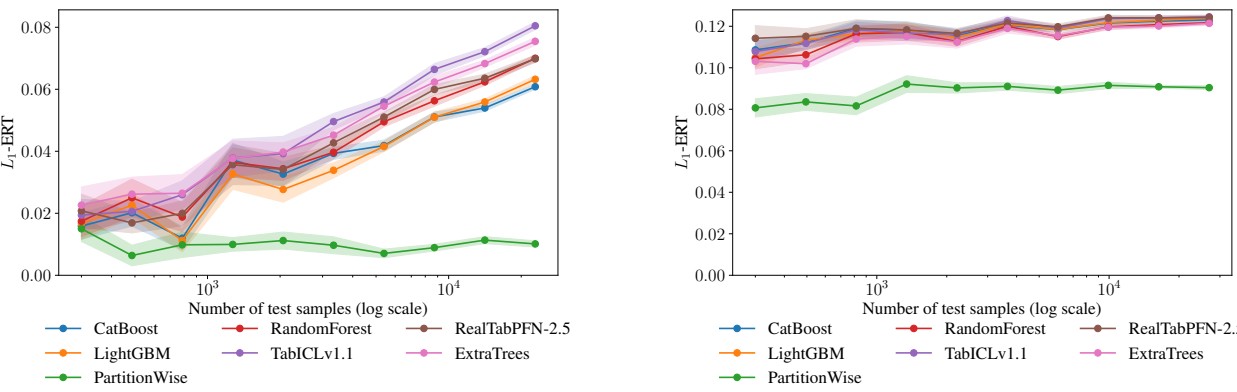

*Figure I.3.* Illustration of the estimation of $L_1$-ERT for different classifiers as a number of available samples. **Left:** physiochemical_protein dataset. **Right:** Diamonds dataset.

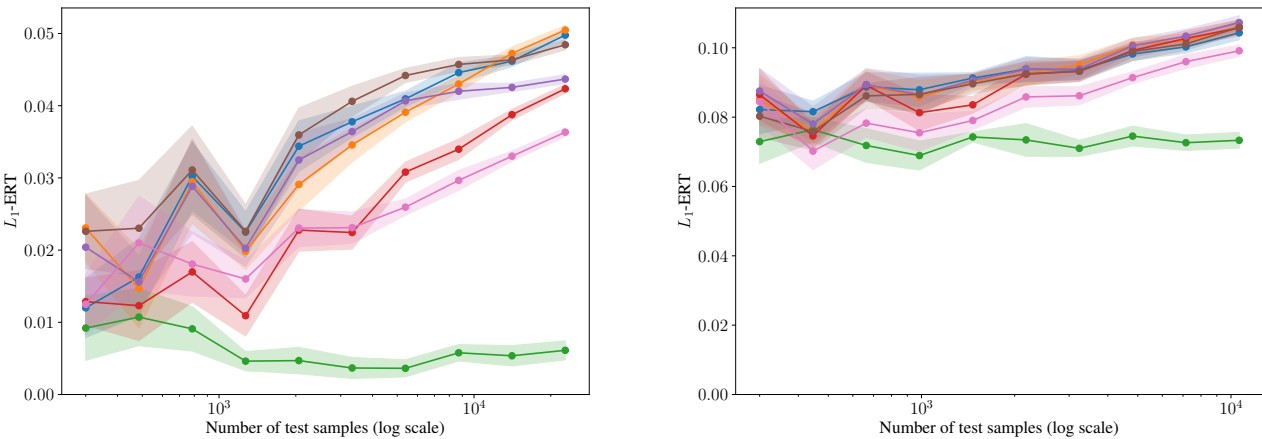

*Figure I.4.* Illustration of the estimation of $L_1$-ERT for different classifiers as a number of sampled data available. Legend shared with Figure I.3. **Left:** Food_Delivery_Time dataset. **Right:** Superconductivity dataset.

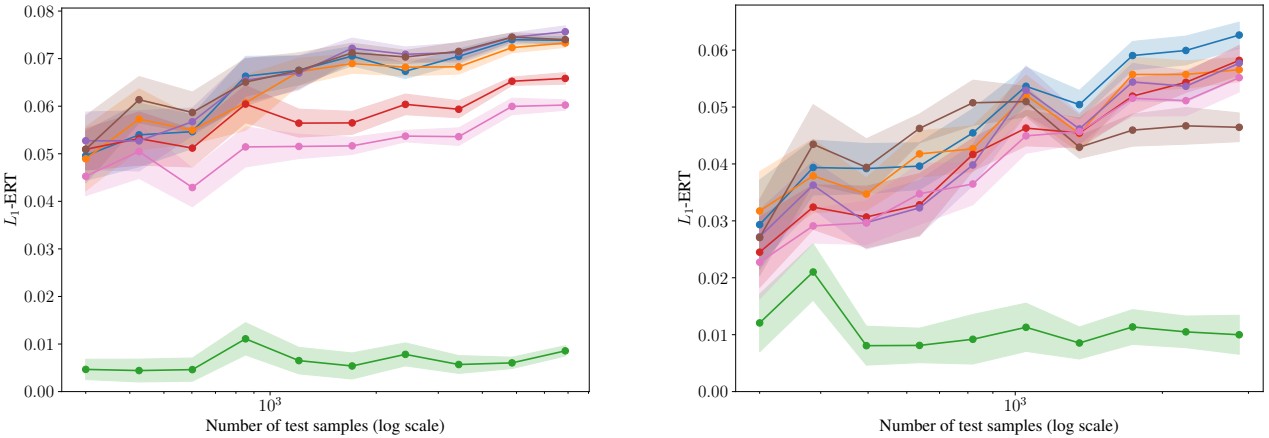

*Figure I.5.* Illustration of the estimation of $L_1$-ERT for different classifiers as a number of sampled data available. Legend shared with Figure I.3. **Left:** ailerons dataset. **Right:** o11 dataset.

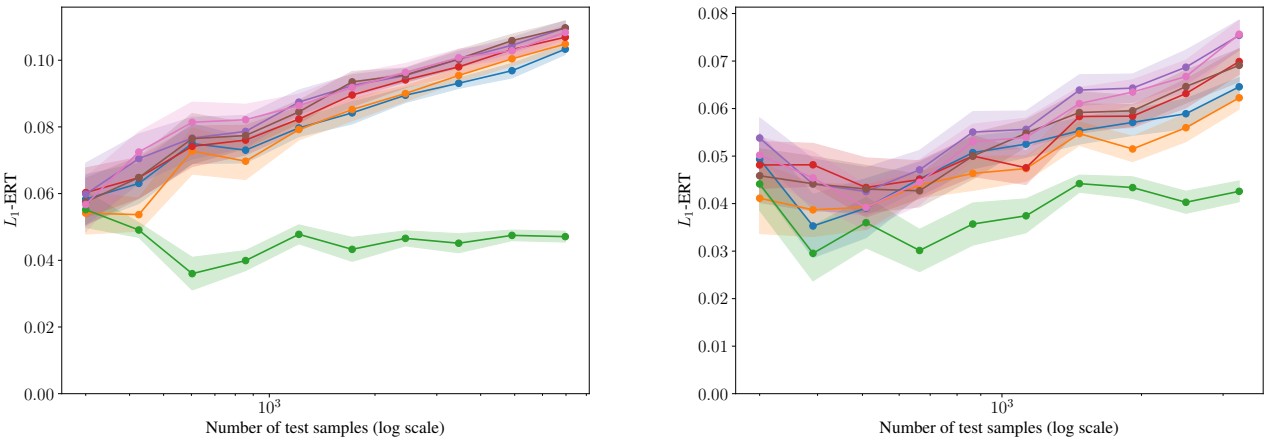

*Figure I.6.* Illustration of the estimation of $L_1$-ERT for different classifiers as a number of sampled data available. Legend shared with Figure I.3. **Left:** miami2016 dataset. **Right:** winequality dataset.

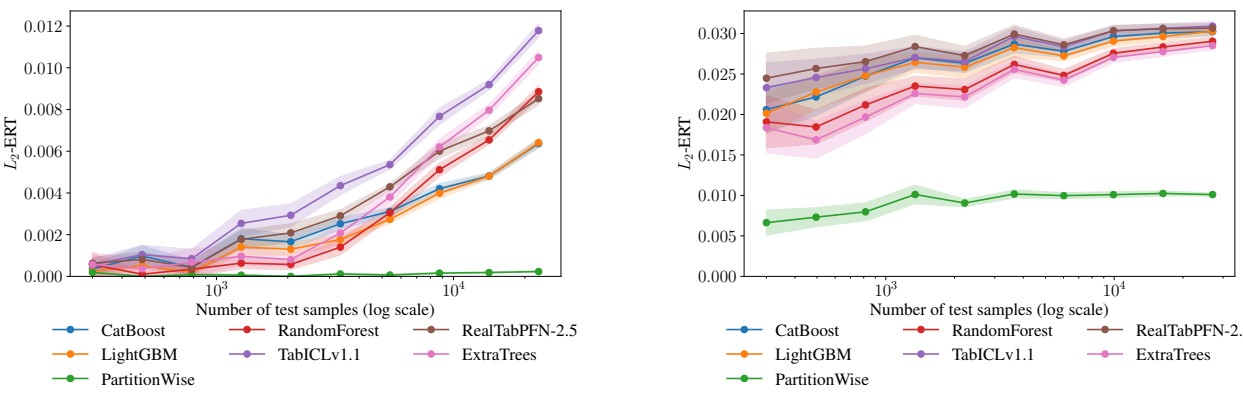

*Figure I.7.* Illustration of the estimation of $L_2$-ERT for different classifiers as a number of sampled data available. **Left:** physiochemical_protein dataset **Right:** Diamonds dataset

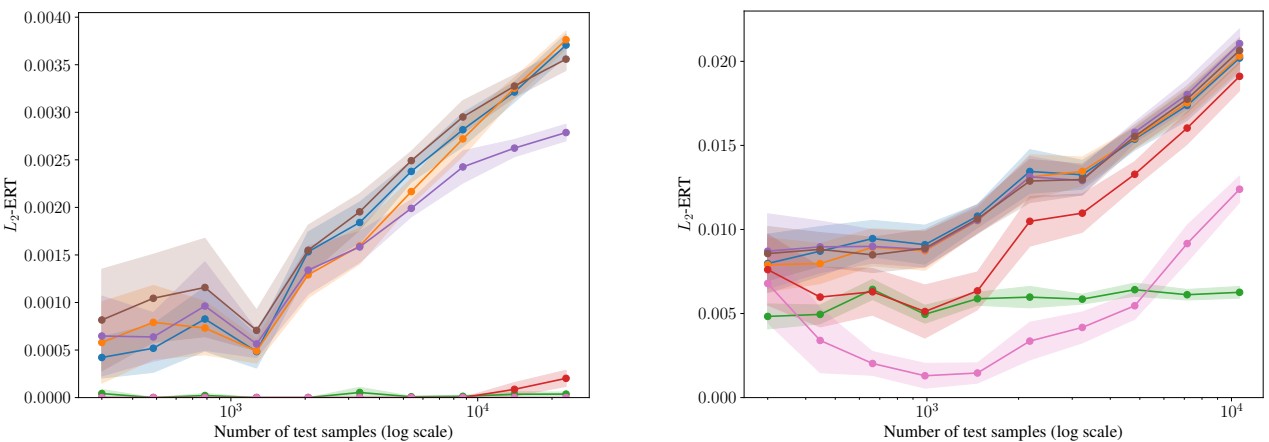

*Figure I.8.* Illustration of the estimation of $L_2$-ERT for different classifiers as a number of sampled data available. Legend shared with Figure I.7. **Left:** Food_Delivery_Time dataset **Right:** Superconductivity dataset

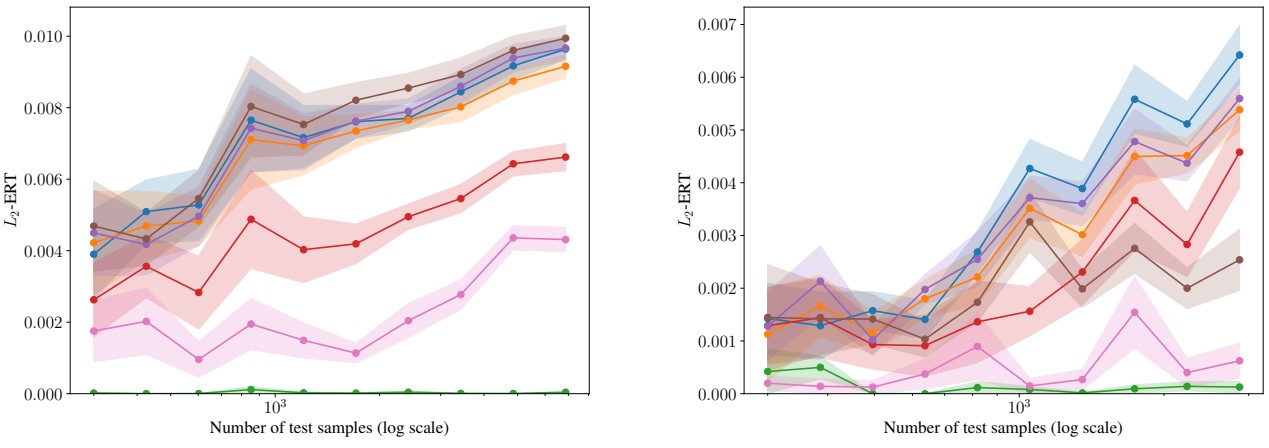

*Figure I.9.* Illustration of the estimation of $L_2$-ERT for different classifiers as a number of sampled data available. Legend shared with Figure I.7. **Left:** ailerons dataset **Right:** o11 dataset

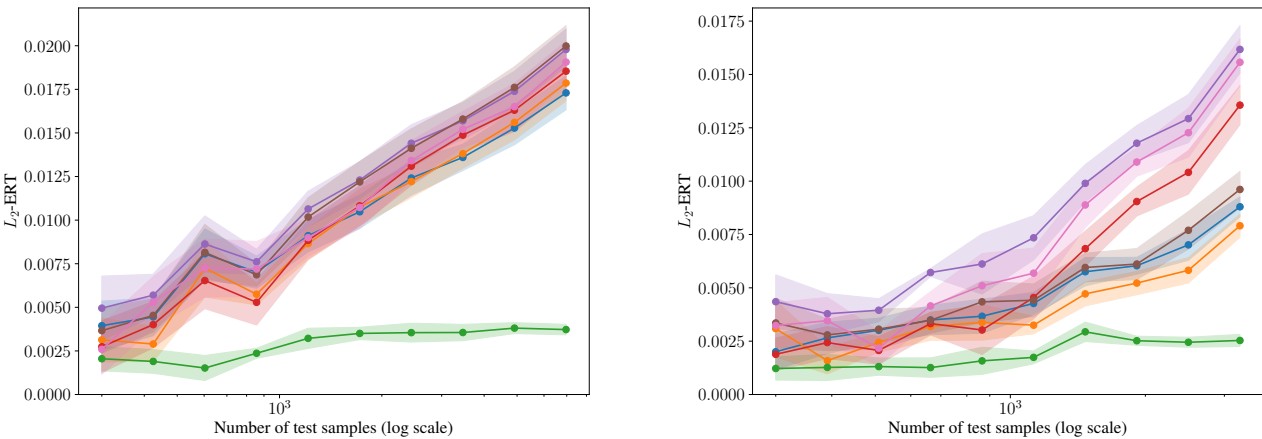

*Figure I.10.* Illustration of the estimation of $L_2$-ERT for different classifiers as a number of sampled data available. Legend shared with Figure I.7. **Left:** miami2016 dataset **Right:** winequality dataset

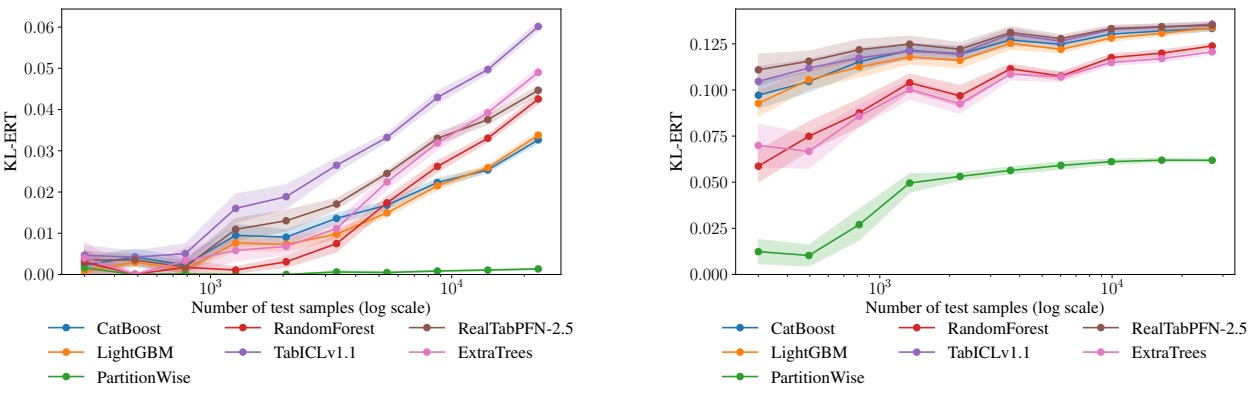

*Figure I.11.* Illustration of the estimation of KL-ERT for different classifiers as a number of sampled data available. **Left:** physiochemical_protein dataset **Right:** Diamonds dataset

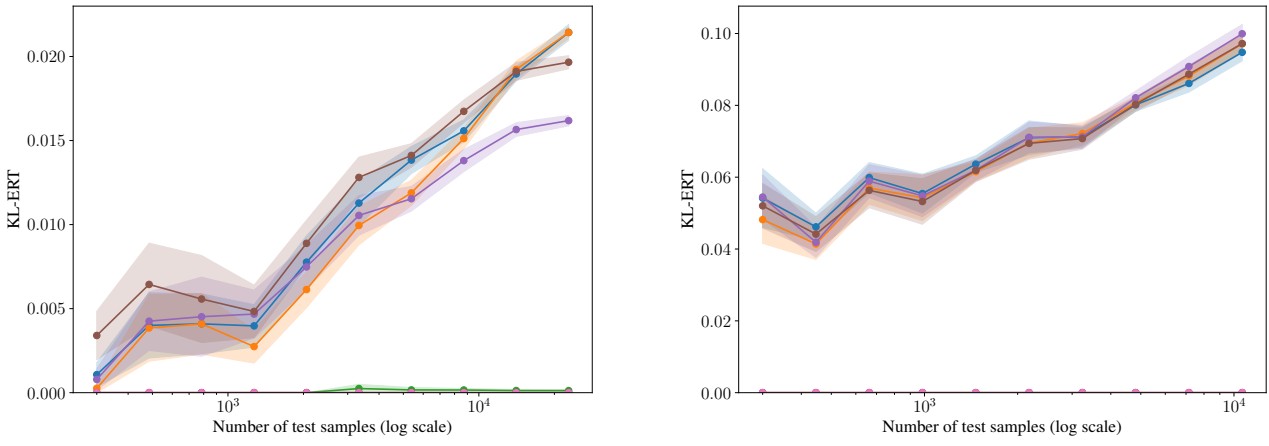

*Figure I.12.* Illustration of the estimation of KL-ERT for different classifiers as a number of sampled data available. Legend shared with Figure I.11. **Left:** Food_Delivery_Time dataset **Right:** Superconductivity dataset

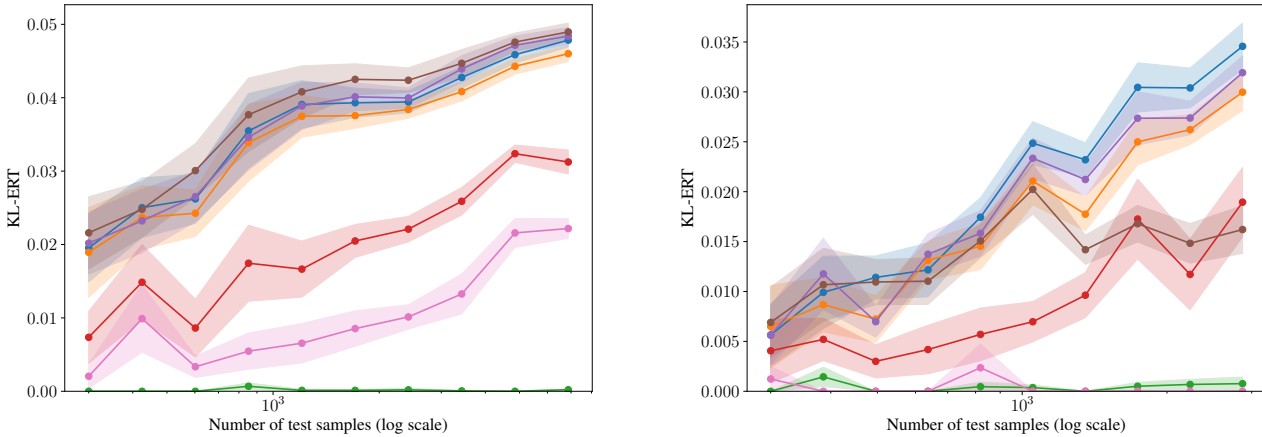

*Figure I.13.* Illustration of the estimation of KL-ERT for different classifiers as a number of sampled data available. Legend shared with Figure I.11. **Left:** ailerons dataset **Right:** o11 dataset

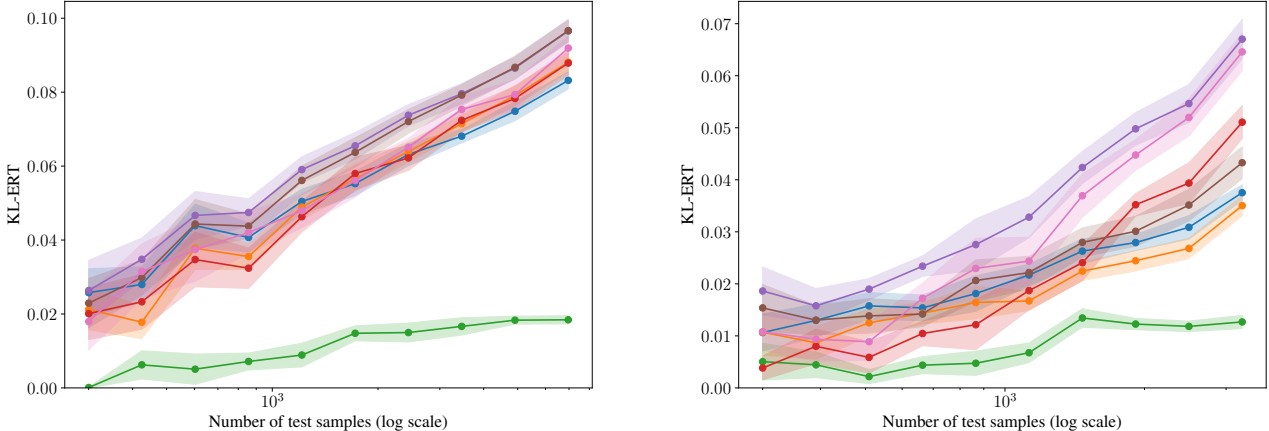

*Figure I.14.* Illustration of the estimation of KL-ERT for different classifiers as a number of sampled data available. Legend shared with Figure I.11. **Left:** miami2016 dataset **Right:** winequality dataset

# J. Real datasets benchmark

## J.1. Regression

**Multivariate results.** We present aggregated results across six datasets, with dataset specific information provided in Appendix I. Each metric is averaged over all datasets, except for the volume for which we averaged the ranks. Results are available in Figures J.1 & J.2 & J.3.

These results, however, must be interpreted with care. As shown in Figure J.3, improvements in conditional coverage often come hand-in-hand with larger prediction sets. For instance, the C-PCP (Dheur et al., 2025) method achieves one of the best conditional coverage across all metrics, but also produces one of the largest prediction intervals. Conversely, methods such as MVCS (Braun et al., 2025a) yield smaller predictive sets at the cost of poorer conditional coverage. This observation underscores a fundamental trade-off: strategies that aggressively minimize prediction volume while maintaining marginal coverage often sacrifice conditional coverage. Understanding and managing this trade-off is crucial for tailoring conformal prediction methods to specific applications.

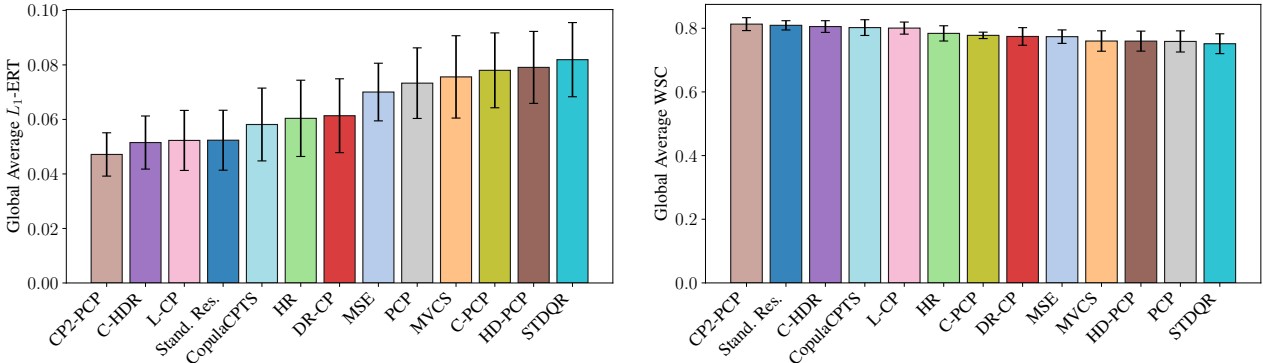

*Figure J.1.* Metric values averaged across all datasets for all methods in multivariate regression. **Left:** $L_1$-ERT (lower is better). **Right:** WSC (closer to 0.9 is better)

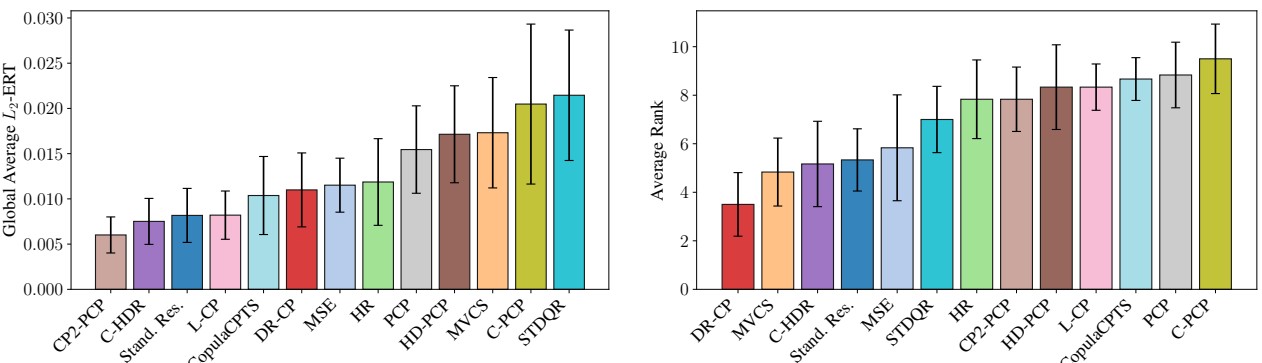

*Figure J.2.* Metric values averaged across all datasets for all methods in multivariate regression. **Left:** $L_2$-ERT (lower is better). **Right:** Normalized set sizes averaged all datasets in multivariate regression, where the normalization is done by dividing each volume by the smallest volume across all methods (smaller is better)

**Uni-variate regression.** We show the same results for the uni-variate datasets, with similar results in Figures J.4 & J.5 & J.6.

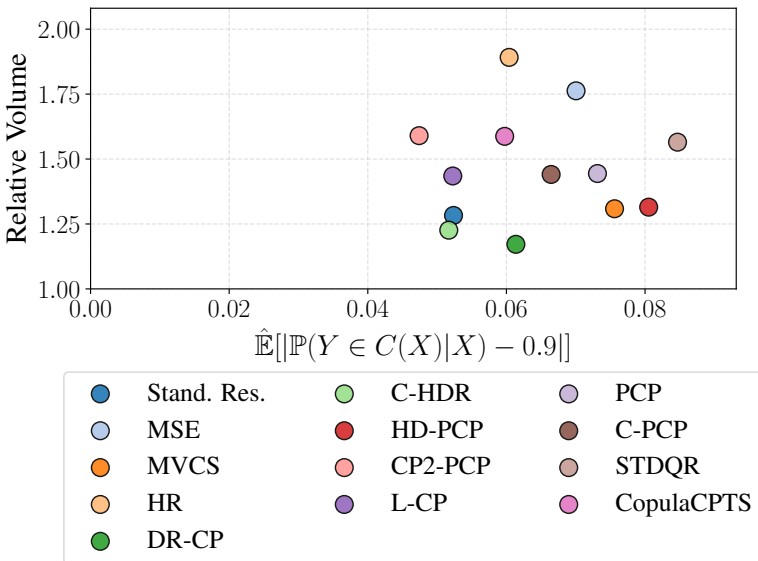

*Figure J.3.* Comparison of conditional coverage deviation and normalized prediction set volumes across various conformal prediction strategies ($\alpha = 0.1$) for multi dimensional responses datasets. Volumes are scaled by the power $1/d$ and normalized by the minimum observed volume. The $x$-axis displays the $L_1$-ERT metric, which estimates the conditional coverage deviation $\mathbb{E}[\|\mathbb{P}(Y \in C(X) \mid X) - (1 - \alpha)\|]$.

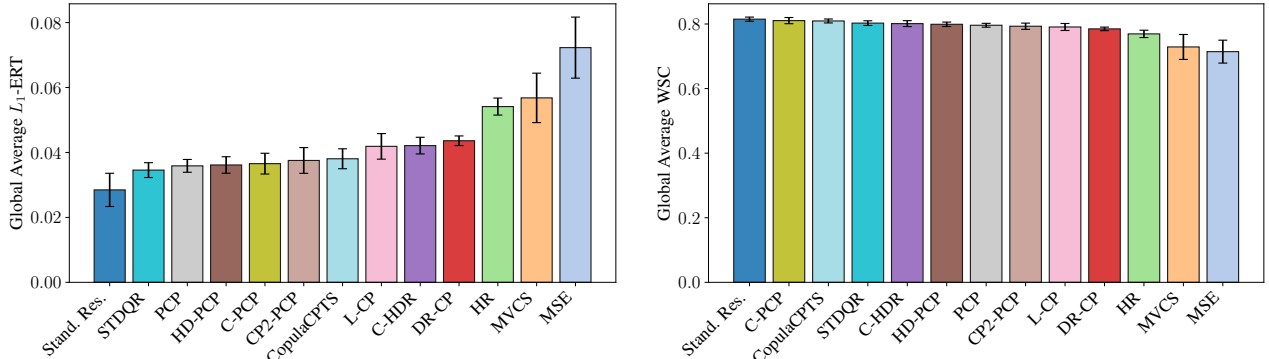

*Figure J.4.* Metric values averaged across all datasets for all methods in uni-variate regression. **Left:** $L_1$-ERT (lower is better). **Right:** WSC (closer to 0.9 is better)

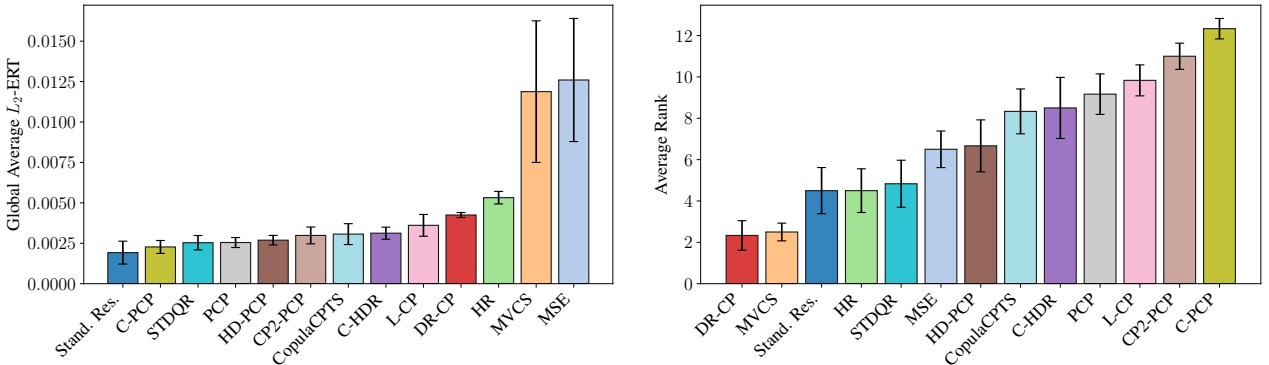

*Figure J.5.* Metric values averaged across all datasets for all methods in univariate regression. **Left:** $L_2$-ERT (lower is better). **Right:** Normalized set sizes averaged all datasets in multivariate regression, where the normalization is done by dividing each volume by the smallest volume across all methods (smaller is better)

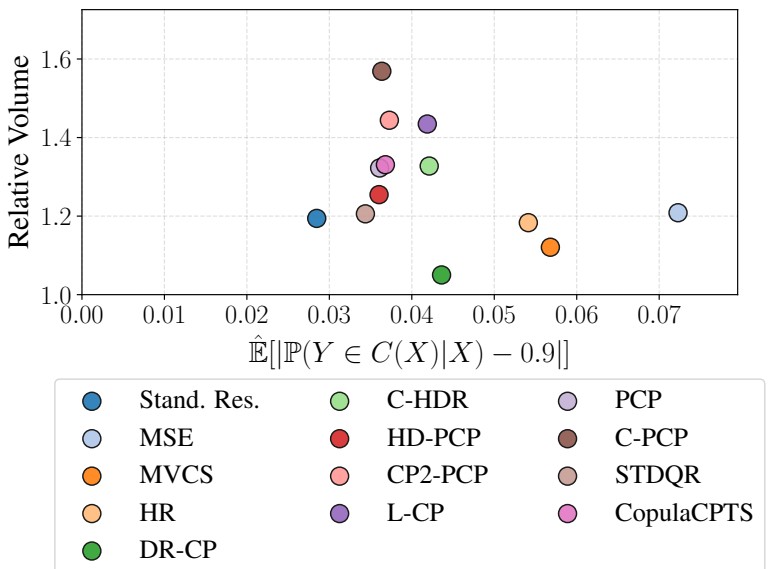

*Figure J.6.* Comparison of conditional coverage deviation and normalized prediction set volumes across various conformal prediction strategies ($\alpha = 0.1$) for uni-variate responses datasets. Volumes are scaled by the power $1/d$ and normalized by the minimum observed volume. The $x$-axis displays the $L_1$-ERT metric, which estimates the conditional coverage deviation $\mathbb{E}[|\mathbb{P}(Y \in C(X) \mid X) - (1 - \alpha)|]$.

## J.2. Classification

In classification, most of the strategies are commonly tailored to specific problems such as long-tailed classification, so we choose to only compare the two most used conformal prediction strategies for classification, given a predictive model that returns probability estimates $\hat{f}(X) \in \Delta_d$. The first one is the negative likelihood prediction (Sadinle et al., 2019) and uses the score $S(X, Y) = -p(X)_Y$. The second one (Romano et al., 2020b; Angelopoulos et al., 2021) uses the cumulative likelihood scores. We first define the permutation $\pi(x)$ of $\{1, \ldots, K\}$ that sorts the probabilities in decreasing order, i.e.,

$$\hat{f}_{\pi_1(x)}(x) \geq \hat{f}_{\pi_2(x)}(x) \geq \cdots \geq \hat{f}_{\pi_K(x)}(x).$$

Then, the score function is defined as:

$$S(X, Y) = \sum_{j=1}^{\pi_k(X)} \hat{f}_{\pi_j(X)}(X), \quad \text{where } Y = \pi_k(X).$$

For MNIST, FashionMNIST, and CIFAR, we trained a CNN composed of two convolutional layers followed by max pooling, then two fully connected layers with dropout and ReLU activations. The model was trained with cross entropy loss to learn $\hat{f}$. We used early stopping when the accuracy fell below $1 - \alpha$, since otherwise both conformal strategies tend to produce many empty sets, which would make the results uninformative.

For the CIFAR100 experiment, we trained a ResNet model with cross-entropy loss to learn $\hat{f}$. To learn the classifier for ERT, we re-used this pretrained model, but replaced its final layer with a new one. This avoided the cost of learning a large feature space from scratch.

We report the ERT values in Table J.1. For the classification problem, both strategies remain far from conditional. In general, we believe that calibrating the predictors leads to better conditional coverage. Interestingly for CIFAR100, the $L_1$-ERT and the KL-ERT lead to two different conclusions: the former suggests that the likelihood strategy is more conditional than the cumulative one, while the latter suggests the opposite. We attribute this discrepancy to the larger number of empty predictive sets produced by the likelihood strategy, for which the conditional coverage equals zero, that are weighted differently by the KL than the $L_1$. This is supported by the analysis of under-coverage and over-coverage. This situation happens more frequently under the likelihood strategy. As a consequence, this strategy yields a larger value of $KL_-$-ERT than $KL_+$-ERT, since the KL divergence assigns more weight to such extreme situations.

*Table J.1.* ERT scores obtained for the classification problems.

| Dataset | Method | $L_1$-ERT | KL-ERT | $KL_+$-ERT | $KL_-$-ERT |
|---|---|---|---|---|---|
| CIFAR10 | cumulative | $0.072_{0.005}$ | $-0.017_{0.008}$ | $-0.030_{0.005}$ | $0.012_{0.006}$ |
| | likelihood | $0.016_{0.002}$ | $0.028_{0.006}$ | $0.007_{0.001}$ | $0.022_{0.007}$ |
| CIFAR100 | cumulative | $0.041_{0.005}$ | $0.191_{0.024}$ | $0.016_{0.008}$ | $0.175_{0.026}$ |
| | likelihood | $0.007_{0.003}$ | $0.409_{0.025}$ | $0.085_{0.022}$ | $0.323_{0.020}$ |
| FashionMNIST | cumulative | $0.165_{0.004}$ | $-0.260_{0.014}$ | $-0.185_{0.010}$ | $-0.075_{0.004}$ |
| | likelihood | $0.098_{0.005}$ | $-0.068_{0.008}$ | $-0.042_{0.006}$ | $-0.026_{0.005}$ |
| MNIST | cumulative | $0.150_{0.006}$ | $-0.216_{0.017}$ | $-0.159_{0.012}$ | $-0.057_{0.006}$ |
| | likelihood | $0.145_{0.003}$ | $-0.187_{0.007}$ | $-0.128_{0.005}$ | $-0.059_{0.002}$ |

## J.3. Results per datasets

**Uni-variate regression.**

| Dataset | Method | $L_1$-ERT | $L_2$-ERT | WSC | Size | Time/1k |
|---------|--------|-----------|-----------|-----|------|---------|
| ailerons | Stand. Res. | $\mathbf{0.0179_{0.0063}}$ | $\mathbf{0.0004_{0.0005}}$ | $\mathbf{0.823_{0.010}}$ | $1.35_{0.02}$ | $0.005_{0.001}$ |
| | MSE | $0.0455_{0.0048}$ | $0.0034_{0.0006}$ | $0.786_{0.015}$ | $1.39_{0.04}$ | $0.209_{0.050}$ |
| | MVCS | $\underline{0.0311_{0.0049}}$ | $\underline{0.0015_{0.0007}}$ | $0.809_{0.011}$ | $1.37_{0.03}$ | $0.050_{0.010}$ |
| | HR | $0.0500_{0.0053}$ | $0.0038_{0.0010}$ | $0.787_{0.022}$ | $1.50_{0.04}$ | $\mathbf{0.001_{0.000}}$ |
| | DR-CP | $0.0494_{0.0139}$ | $0.0041_{0.0023}$ | $0.785_{0.030}$ | $\mathbf{1.21_{0.37}}$ | $0.005_{0.001}$ |
| | C-HDR | $0.0450_{0.0157}$ | $0.0033_{0.0015}$ | $0.803_{0.022}$ | $1.45_{0.34}$ | $28.300_{22.902}$ |
| | HD-PCP | $0.0405_{0.0146}$ | $0.0028_{0.0020}$ | $0.789_{0.033}$ | $1.51_{0.17}$ | $36.439_{26.716}$ |
| | CP2-PCP | $0.0468_{0.0154}$ | $0.0041_{0.0017}$ | $0.794_{0.050}$ | $1.61_{0.14}$ | $65.635_{42.957}$ |
| | L-CP | $0.0396_{0.0160}$ | $0.0029_{0.0020}$ | $0.796_{0.022}$ | $1.75_{0.54}$ | $\underline{0.002_{0.000}}$ |
| | PCP | $0.0426_{0.0182}$ | $0.0033_{0.0024}$ | $0.791_{0.036}$ | $1.57_{0.19}$ | $27.889_{23.093}$ |
| | C-PCP | $0.0390_{0.0164}$ | $0.0025_{0.0015}$ | $\underline{0.811_{0.031}}$ | $1.87_{0.21}$ | $85.880_{69.424}$ |
| | STDQR | $0.0319_{0.0142}$ | $0.0021_{0.0014}$ | $0.804_{0.023}$ | $1.52_{0.21}$ | $31.947_{21.252}$ |
| | CopulaCPTS | $0.0327_{0.0129}$ | $0.0022_{0.0011}$ | $0.811_{0.032}$ | $1.69_{0.55}$ | $36.448_{27.553}$ |
| bank8FM | Stand. Res. | $\mathbf{0.0138_{0.0095}}$ | $\mathbf{0.0006_{0.0006}}$ | $\underline{0.800_{0.013}}$ | $0.91_{0.05}$ | $0.006_{0.001}$ |
| | MSE | $0.0975_{0.0110}$ | $0.0279_{0.0044}$ | $0.572_{0.034}$ | $0.81_{0.03}$ | $0.098_{0.020}$ |
| | MVCS | $0.0589_{0.0113}$ | $0.0099_{0.0035}$ | $0.700_{0.029}$ | $\mathbf{0.71_{0.03}}$ | $0.049_{0.009}$ |
| | HR | $0.0528_{0.0107}$ | $0.0054_{0.0017}$ | $0.726_{0.023}$ | $0.80_{0.03}$ | $\mathbf{0.003_{0.000}}$ |
| | DR-CP | $0.0450_{0.0084}$ | $0.0044_{0.0018}$ | $0.780_{0.029}$ | $\underline{0.77_{0.06}}$ | $0.007_{0.001}$ |
| | C-HDR | $0.0430_{0.0065}$ | $0.0032_{0.0016}$ | $0.796_{0.028}$ | $1.09_{0.16}$ | $12.659_{3.011}$ |
| | HD-PCP | $0.0355_{0.0092}$ | $\underline{0.0023_{0.0016}}$ | $0.795_{0.022}$ | $0.81_{0.06}$ | $17.214_{3.679}$ |
| | CP2-PCP | $0.0360_{0.0105}$ | $0.0030_{0.0021}$ | $0.786_{0.030}$ | $1.02_{0.11}$ | $33.233_{5.569}$ |
| | L-CP | $0.0365_{0.0135}$ | $0.0033_{0.0024}$ | $0.780_{0.029}$ | $1.05_{0.09}$ | $\underline{0.003_{0.001}}$ |
| | PCP | $0.0386_{0.0073}$ | $0.0035_{0.0019}$ | $0.787_{0.036}$ | $0.85_{0.07}$ | $12.287_{2.979}$ |
| | C-PCP | $0.0419_{0.0106}$ | $0.0031_{0.0017}$ | $\mathbf{0.806_{0.029}}$ | $1.19_{0.19}$ | $37.293_{9.280}$ |
| | STDQR | $\underline{0.0346_{0.0107}}$ | $0.0026_{0.0019}$ | $0.790_{0.021}$ | $0.83_{0.05}$ | $18.579_{3.534}$ |
| | CopulaCPTS | $0.0426_{0.0153}$ | $0.0038_{0.0034}$ | $0.793_{0.066}$ | $0.89_{0.08}$ | $16.706_{3.119}$ |
| cpu.act | Stand. Res. | $0.0368_{0.0099}$ | $0.0027_{0.0015}$ | $\underline{0.798_{0.023}}$ | $1.13_{0.08}$ | $0.006_{0.001}$ |
| | MSE | $0.0964_{0.0105}$ | $0.0194_{0.0045}$ | $0.675_{0.032}$ | $1.14_{0.07}$ | $0.152_{0.050}$ |
| | MVCS | $0.0855_{0.0050}$ | $0.0311_{0.0055}$ | $0.574_{0.053}$ | $\underline{1.05_{0.05}}$ | $0.053_{0.012}$ |
| | HR | $0.0559_{0.0044}$ | $0.0057_{0.0012}$ | $0.769_{0.015}$ | $1.07_{0.03}$ | $\mathbf{0.002_{0.000}}$ |
| | DR-CP | $0.0394_{0.0075}$ | $0.0042_{0.0033}$ | $0.772_{0.038}$ | $\mathbf{0.97_{0.13}}$ | $0.007_{0.001}$ |
| | C-HDR | $0.0473_{0.0108}$ | $0.0039_{0.0018}$ | $0.766_{0.028}$ | $1.46_{0.17}$ | $27.366_{7.185}$ |
| | HD-PCP | $\mathbf{0.0299_{0.0098}}$ | $0.0022_{0.0016}$ | $0.795_{0.025}$ | $1.09_{0.11}$ | $34.721_{7.917}$ |
| | CP2-PCP | $0.0459_{0.0090}$ | $0.0040_{0.0021}$ | $0.762_{0.039}$ | $1.68_{1.20}$ | $66.768_{14.695}$ |
| | L-CP | $0.0508_{0.0109}$ | $0.0048_{0.0025}$ | $0.756_{0.031}$ | $1.47_{0.08}$ | $\underline{0.004_{0.001}}$ |
| | PCP | $0.0307_{0.0058}$ | $\mathbf{0.0015_{0.0006}}$ | $0.793_{0.020}$ | $1.14_{0.11}$ | $27.063_{7.232}$ |
| | C-PCP | $0.0433_{0.0092}$ | $0.0032_{0.0010}$ | $0.776_{0.019}$ | $1.62_{0.06}$ | $77.444_{17.833}$ |
| | STDQR | $\underline{0.0305_{0.0079}}$ | $\underline{0.0016_{0.0010}}$ | $0.797_{0.018}$ | $1.11_{0.12}$ | $35.379_{6.900}$ |
| | CopulaCPTS | $0.0370_{0.0240}$ | $0.0026_{0.0028}$ | $\mathbf{0.815_{0.061}}$ | $1.17_{0.10}$ | $33.708_{5.958}$ |

| Dataset | Method | $L_1$-ERT | $L_2$-ERT | WSC | Size | Time/1k |
|---|---|---|---|---|---|---|
| house_8L | Stand. Res. | $0.0223_{0.0032}$ | $0.0008_{0.0004}$ | $0.839_{0.009}$ | $1.62_{0.04}$ | $0.005_{0.001}$ |
| | MSE | $0.0580_{0.0026}$ | $0.0067_{0.0011}$ | $0.781_{0.020}$ | $1.63_{0.03}$ | $0.201_{0.048}$ |
| | MVCS | $0.0436_{0.0052}$ | $0.0051_{0.0008}$ | $0.813_{0.012}$ | $1.50_{0.03}$ | $0.055_{0.026}$ |
| | HR | $0.0533_{0.0039}$ | $0.0051_{0.0009}$ | $0.798_{0.018}$ | $1.64_{0.04}$ | $\mathbf{0.002_{0.000}}$ |
| | DR-CP | $0.0404_{0.0039}$ | $0.0046_{0.0009}$ | $0.806_{0.019}$ | $\mathbf{1.35_{0.12}}$ | $0.006_{0.001}$ |
| | C-HDR | $0.0296_{0.0063}$ | $0.0014_{0.0007}$ | $0.836_{0.017}$ | $1.50_{0.11}$ | $54.250_{22.089}$ |
| | HD-PCP | $0.0283_{0.0063}$ | $0.0018_{0.0007}$ | $0.829_{0.015}$ | $2.92_{2.67}$ | $72.573_{28.698}$ |
| | CP2-PCP | $\mathbf{0.0200_{0.0049}}$ | $\underline{0.0006_{0.0004}}$ | $0.835_{0.011}$ | $1.76_{0.09}$ | $139.272_{51.333}$ |
| | L-CP | $0.0275_{0.0093}$ | $0.0013_{0.0010}$ | $0.837_{0.020}$ | $1.64_{0.08}$ | $\underline{0.002_{0.001}}$ |
| | PCP | $0.0324_{0.0037}$ | $0.0023_{0.0005}$ | $0.822_{0.013}$ | $3.06_{2.66}$ | $53.370_{21.916}$ |
| | C-PCP | $\underline{0.0218_{0.0037}}$ | $\mathbf{0.0005_{0.0004}}$ | $\mathbf{0.849_{0.016}}$ | $1.95_{0.31}$ | $162.786_{63.893}$ |
| | STDQR | $0.0281_{0.0048}$ | $0.0016_{0.0007}$ | $0.835_{0.014}$ | $\underline{1.49_{0.05}}$ | $65.639_{28.135}$ |
| | CopulaCPTS | $0.0273_{0.0055}$ | $0.0014_{0.0007}$ | $0.832_{0.021}$ | $2.51_{2.54}$ | $70.871_{27.325}$ |
| miami | Stand. Res. | $\mathbf{0.0334_{0.0061}}$ | $\mathbf{0.0020_{0.0010}}$ | $0.823_{0.015}$ | $\underline{0.87_{0.03}}$ | $0.005_{0.001}$ |
| | MSE | $0.0837_{0.0048}$ | $0.0112_{0.0014}$ | $0.685_{0.026}$ | $0.98_{0.05}$ | $0.131_{0.026}$ |
| | MVCS | $0.0655_{0.0065}$ | $0.0166_{0.0017}$ | $0.684_{0.023}$ | $0.88_{0.04}$ | $0.048_{0.001}$ |
| | HR | $0.0657_{0.0071}$ | $0.0067_{0.0008}$ | $0.747_{0.014}$ | $0.89_{0.03}$ | $\mathbf{0.002_{0.000}}$ |
| | DR-CP | $0.0426_{0.0097}$ | $0.0036_{0.0012}$ | $0.771_{0.022}$ | $0.88_{0.03}$ | $0.005_{0.001}$ |
| | C-HDR | $0.0442_{0.0055}$ | $0.0030_{0.0006}$ | $0.803_{0.016}$ | $1.04_{0.04}$ | $11.236_{2.096}$ |
| | HD-PCP | $0.0445_{0.0071}$ | $0.0036_{0.0010}$ | $0.780_{0.026}$ | $0.88_{0.04}$ | $15.867_{2.345}$ |
| | CP2-PCP | $0.0395_{0.0063}$ | $0.0029_{0.0013}$ | $0.787_{0.025}$ | $1.05_{0.04}$ | $29.755_{4.660}$ |
| | L-CP | $0.0430_{0.0078}$ | $0.0033_{0.0011}$ | $0.785_{0.026}$ | $0.97_{0.04}$ | $\underline{0.002_{0.000}}$ |
| | PCP | $0.0390_{0.0061}$ | $0.0028_{0.0009}$ | $0.781_{0.023}$ | $0.97_{0.03}$ | $10.868_{2.037}$ |
| | C-PCP | $\underline{0.0389_{0.0073}}$ | $\underline{0.0021_{0.0009}}$ | $0.808_{0.024}$ | $1.14_{0.05}$ | $32.202_{6.722}$ |
| | STDQR | $0.0398_{0.0066}$ | $0.0030_{0.0004}$ | $0.785_{0.019}$ | $\mathbf{0.87_{0.03}}$ | $14.499_{1.771}$ |
| | CopulaCPTS | $0.0401_{0.0059}$ | $0.0025_{0.0009}$ | $\underline{0.813_{0.023}}$ | $0.92_{0.05}$ | $15.120_{3.060}$ |
| sulfur | Stand. Res. | $0.0466_{0.0044}$ | $0.0050_{0.0017}$ | $\underline{0.807_{0.016}}$ | $1.43_{0.04}$ | $0.006_{0.001}$ |
| | MSE | $0.0526_{0.0095}$ | $0.0069_{0.0030}$ | $0.788_{0.023}$ | $1.49_{0.04}$ | $0.196_{0.023}$ |
| | MVCS | $0.0563_{0.0047}$ | $0.0071_{0.0013}$ | $0.795_{0.010}$ | $\underline{1.43_{0.04}}$ | $0.048_{0.001}$ |
| | HR | $0.0471_{0.0070}$ | $0.0052_{0.0022}$ | $0.790_{0.015}$ | $\mathbf{1.36_{0.03}}$ | $\mathbf{0.002_{0.000}}$ |
| | DR-CP | $0.0448_{0.0071}$ | $0.0046_{0.0024}$ | $0.795_{0.018}$ | $1.48_{0.04}$ | $0.006_{0.001}$ |
| | C-HDR | $0.0436_{0.0085}$ | $0.0039_{0.0011}$ | $0.804_{0.024}$ | $1.58_{0.04}$ | $13.660_{1.837}$ |
| | HD-PCP | $0.0381_{0.0063}$ | $0.0035_{0.0015}$ | $0.805_{0.018}$ | $1.51_{0.05}$ | $17.520_{2.751}$ |
| | CP2-PCP | $0.0369_{0.0091}$ | $0.0035_{0.0013}$ | $0.794_{0.025}$ | $1.67_{0.05}$ | $32.987_{5.117}$ |
| | L-CP | $0.0538_{0.0078}$ | $0.0061_{0.0015}$ | $0.791_{0.011}$ | $1.52_{0.04}$ | $\underline{0.003_{0.000}}$ |
| | PCP | $\mathbf{0.0319_{0.0060}}$ | $\mathbf{0.0020_{0.0007}}$ | $0.803_{0.017}$ | $1.65_{0.03}$ | $13.102_{1.832}$ |
| | C-PCP | $\underline{0.0343_{0.0109}}$ | $\underline{0.0023_{0.0011}}$ | $\mathbf{0.812_{0.020}}$ | $1.74_{0.05}$ | $39.555_{7.945}$ |
| | STDQR | $0.0425_{0.0065}$ | $0.0045_{0.0015}$ | $0.805_{0.013}$ | $1.50_{0.11}$ | $16.344_{1.878}$ |
| | CopulaCPTS | $0.0486_{0.0112}$ | $0.0059_{0.0039}$ | $0.792_{0.039}$ | $1.45_{0.10}$ | $16.460_{2.794}$ |

**Multivariate regression**

| Dataset | Method | $L_1$-ERT | $L_2$-ERT | WSC | Size | Time/1k |
|---|---|---|---|---|---|---|
| bias | Stand. Res. | $0.0427_{0.0094}$ | $0.0031_{0.0019}$ | $\mathbf{0.803_{0.026}}$ | $1.25_{0.06}$ | $0.005_{0.001}$ |
| | MSE | $0.0529_{0.0061}$ | $0.0047_{0.0015}$ | $0.792_{0.022}$ | $\mathbf{1.11_{0.02}}$ | $0.130_{0.049}$ |
| | MVCS | $0.0515_{0.0064}$ | $0.0045_{0.0017}$ | $0.794_{0.018}$ | $\underline{1.16_{0.02}}$ | $0.039_{0.014}$ |
| | HR | $0.0492_{0.0054}$ | $0.0041_{0.0013}$ | $0.786_{0.028}$ | $1.21_{0.05}$ | $0.006_{0.001}$ |
| | DR-CP | $0.0451_{0.0049}$ | $0.0041_{0.0015}$ | $0.768_{0.018}$ | $1.39_{0.04}$ | $0.010_{0.003}$ |
| | C-HDR | $0.0458_{nan}$ | $0.0047_{nan}$ | $0.755_{nan}$ | $1.58_{nan}$ | $14.883_{nan}$ |
| | HD-PCP | $0.0434_{0.0078}$ | $0.0032_{0.0020}$ | $\underline{0.795_{0.026}}$ | $1.24_{0.03}$ | $16.330_{3.683}$ |
| | CP2-PCP | $0.0528_{0.0070}$ | $0.0053_{0.0015}$ | $0.758_{0.021}$ | $1.38_{0.04}$ | $25.098_{6.455}$ |
| | L-CP | $0.0509_{0.0066}$ | $0.0056_{0.0021}$ | $0.764_{0.026}$ | $1.47_{0.05}$ | $\mathbf{0.003_{0.001}}$ |
| | PCP | $\mathbf{0.0408_{0.0062}}$ | $\mathbf{0.0027_{0.0019}}$ | $0.790_{0.023}$ | $1.30_{0.03}$ | $7.529_{2.876}$ |
| | C-PCP | $0.0482_{0.0025}$ | $0.0037_{0.0010}$ | $0.774_{0.025}$ | $1.48_{0.08}$ | $22.545_{6.448}$ |
| | STDQR | $0.0443_{0.0069}$ | $0.0033_{0.0019}$ | $0.789_{0.028}$ | $1.24_{0.03}$ | $15.160_{3.211}$ |
| | CopulaCPTS | $0.0436_{0.0095}$ | $\underline{0.0030_{0.0018}}$ | $0.789_{0.034}$ | $1.24_{0.03}$ | $12.566_{3.187}$ |
| casp | Stand. Res. | $0.0543_{0.0046}$ | $0.0074_{0.0013}$ | $0.837_{0.010}$ | $1.42_{0.05}$ | $\underline{0.005_{0.002}}$ |
| | MSE | $0.0703_{0.0035}$ | $0.0110_{0.0010}$ | $0.834_{0.004}$ | $1.28_{0.01}$ | $0.176_{0.059}$ |
| | MVCS | $0.0545_{0.0019}$ | $0.0077_{0.0014}$ | $0.844_{0.012}$ | $1.23_{0.05}$ | $0.048_{0.014}$ |
| | HR | $0.0471_{0.0033}$ | $0.0047_{0.0008}$ | $0.842_{0.005}$ | $1.31_{0.03}$ | $0.006_{0.001}$ |
| | DR-CP | $0.0465_{0.0033}$ | $0.0053_{0.0007}$ | $0.843_{0.008}$ | $\underline{1.21_{0.01}}$ | $0.010_{0.002}$ |
| | C-HDR | $\underline{0.0442_{0.0035}}$ | $0.0046_{0.0006}$ | $0.855_{0.010}$ | $\mathbf{1.11_{0.29}}$ | $170.009_{164.271}$ |
| | HD-PCP | $0.0468_{0.0030}$ | $0.0047_{0.0008}$ | $0.840_{0.008}$ | $2.28_{2.41}$ | $318.492_{395.020}$ |
| | CP2-PCP | $\mathbf{0.0401_{0.0061}}$ | $\mathbf{0.0032_{0.0007}}$ | $\mathbf{0.867_{0.023}}$ | $1.92_{1.27}$ | $224.649_{133.714}$ |
| | L-CP | $0.0481_{0.0038}$ | $0.0062_{0.0008}$ | $0.850_{0.007}$ | $1.63_{0.57}$ | $\mathbf{0.002_{0.000}}$ |
| | PCP | $0.0466_{0.0048}$ | $\underline{0.0043_{0.0007}}$ | $0.832_{0.010}$ | $1.48_{0.16}$ | $169.144_{163.409}$ |
| | C-PCP | $0.1154_{0.2193}$ | $0.0583_{0.1561}$ | $0.764_{0.264}$ | $1.45_{0.03}$ | $311.180_{294.205}$ |
| | STDQR | $0.1086_{0.1960}$ | $0.0491_{0.1403}$ | $0.764_{0.245}$ | $2.09_{1.83}$ | $73.434_{102.011}$ |
| | CopulaCPTS | $0.0525_{0.0181}$ | $0.0061_{0.0018}$ | $\underline{0.859_{0.040}}$ | $1.88_{1.14}$ | $167.442_{187.060}$ |
| house | Stand. Res. | $0.0528_{0.0076}$ | $0.0054_{0.0019}$ | $\underline{0.809_{0.014}}$ | $\mathbf{1.01_{0.04}}$ | $0.004_{0.001}$ |
| | MSE | $0.0848_{0.0046}$ | $0.0142_{0.0015}$ | $0.720_{0.019}$ | $1.18_{0.02}$ | $0.137_{0.036}$ |
| | MVCS | $0.0989_{0.0057}$ | $0.0224_{0.0031}$ | $0.690_{0.034}$ | $1.28_{0.05}$ | $0.027_{0.012}$ |
| | HR | $0.0692_{0.0056}$ | $0.0097_{0.0014}$ | $0.746_{0.017}$ | $1.28_{0.02}$ | $0.005_{0.001}$ |
| | DR-CP | $0.0643_{0.0059}$ | $0.0093_{0.0015}$ | $0.766_{0.016}$ | $\underline{1.04_{0.04}}$ | $0.007_{0.002}$ |
| | C-HDR | $0.0495_{0.0058}$ | $0.0052_{0.0013}$ | $0.804_{0.016}$ | $1.15_{0.05}$ | $16.772_{6.236}$ |
| | HD-PCP | $0.0624_{0.0047}$ | $0.0081_{0.0018}$ | $0.787_{0.012}$ | $1.82_{2.06}$ | $21.125_{6.773}$ |
| | CP2-PCP | $\underline{0.0444_{0.0073}}$ | $\underline{0.0041_{0.0012}}$ | $0.808_{0.012}$ | $1.22_{0.04}$ | $41.761_{13.457}$ |
| | L-CP | $0.0497_{0.0064}$ | $0.0053_{0.0012}$ | $0.800_{0.015}$ | $1.29_{0.07}$ | $\mathbf{0.001_{0.000}}$ |
| | PCP | $0.0628_{0.0081}$ | $0.0084_{0.0022}$ | $0.774_{0.017}$ | $1.89_{2.05}$ | $15.975_{6.212}$ |
| | C-PCP | $\mathbf{0.0442_{0.0069}}$ | $\mathbf{0.0038_{0.0012}}$ | $\mathbf{0.813_{0.017}}$ | $2.05_{2.38}$ | $50.679_{24.782}$ |
| | STDQR | $0.0607_{0.0047}$ | $0.0078_{0.0016}$ | $0.784_{0.013}$ | $1.59_{1.30}$ | $20.781_{8.768}$ |
| | CopulaCPTS | $0.0546_{0.0065}$ | $0.0066_{0.0025}$ | $0.799_{0.028}$ | $1.28_{0.24}$ | $19.358_{6.187}$ |

| Dataset | Method | $L_1$-ERT | $L_2$-ERT | WSC | Size | Time/1k |
|---------|--------|-----------|-----------|-----|------|---------|
| rf1 | Stand. Res. | $0.0743_{0.0092}$ | $0.0137_{0.0038}$ | $0.773_{0.016}$ | $\underline{0.31}_{0.01}$ | $0.006_{0.001}$ |
| | MSE | $0.0954_{0.0085}$ | $0.0189_{0.0051}$ | $0.703_{0.029}$ | $0.95_{0.00}$ | $0.178_{0.034}$ |
| | MVCS | $0.1045_{0.0079}$ | $0.0294_{0.0036}$ | $0.645_{0.035}$ | $\mathbf{0.30_{0.01}}$ | $0.049_{0.001}$ |
| | HR | $0.0904_{0.0124}$ | $0.0241_{0.0074}$ | $0.719_{0.042}$ | $0.62_{0.03}$ | $0.025_{0.001}$ |
| | DR-CP | $0.1052_{0.0088}$ | $0.0248_{0.0057}$ | $0.655_{0.032}$ | $0.37_{0.01}$ | $0.034_{0.002}$ |
| | C-HDR | $0.0785_{0.0071}$ | $0.0145_{0.0047}$ | $0.765_{0.028}$ | $0.39_{0.02}$ | $21.148_{2.452}$ |
| | HD-PCP | $0.1194_{0.0089}$ | $0.0308_{0.0037}$ | $0.614_{0.037}$ | $0.41_{0.04}$ | $26.383_{4.228}$ |
| | CP2-PCP | $\mathbf{0.0631_{0.0062}}$ | $\mathbf{0.0096_{0.0026}}$ | $\underline{0.778}_{0.023}$ | $0.39_{0.01}$ | $47.723_{5.485}$ |
| | L-CP | $0.0813_{0.0082}$ | $0.0153_{0.0045}$ | $0.751_{0.034}$ | $0.40_{0.01}$ | $\mathbf{0.003_{0.000}}$ |
| | PCP | $0.1208_{0.0065}$ | $0.0308_{0.0036}$ | $0.599_{0.036}$ | $0.41_{0.04}$ | $18.907_{2.283}$ |
| | C-PCP | $\underline{0.0663}_{0.0113}$ | $\underline{0.0103}_{0.0044}$ | $\mathbf{0.787_{0.025}}$ | $0.39_{0.01}$ | $55.434_{6.881}$ |
| | STDQR | $0.1210_{0.0088}$ | $0.0321_{0.0039}$ | $0.601_{0.041}$ | $0.39_{0.02}$ | $29.116_{3.245}$ |
| | CopulaCPTS | $0.0993_{0.0083}$ | $0.0269_{0.0063}$ | $0.703_{0.031}$ | $0.40_{0.05}$ | $24.389_{2.094}$ |
| rf2 | Stand. Res. | $0.0834_{0.0055}$ | $0.0197_{0.0041}$ | $0.771_{0.021}$ | $0.38_{0.02}$ | $\underline{0.005}_{0.001}$ |
| | MSE | $0.0894_{0.0064}$ | $0.0189_{0.0046}$ | $0.778_{0.019}$ | $0.96_{0.01}$ | $0.207_{0.045}$ |
| | MVCS | $0.1191_{0.0067}$ | $0.0383_{0.0030}$ | $0.759_{0.026}$ | $0.47_{0.02}$ | $0.033_{0.013}$ |
| | HR | $0.1006_{0.0134}$ | $0.0286_{0.0081}$ | $0.744_{0.062}$ | $1.20_{0.12}$ | $0.036_{0.011}$ |
| | DR-CP | $0.0921_{0.0084}$ | $0.0219_{0.0042}$ | $0.775_{0.017}$ | $\underline{0.34}_{0.02}$ | $0.024_{0.006}$ |
| | C-HDR | $0.0768_{0.0061}$ | $0.0158_{0.0036}$ | $\mathbf{0.790_{0.019}}$ | $0.35_{0.01}$ | $32.829_{3.846}$ |
| | HD-PCP | $0.1047_{0.0084}$ | $0.0266_{0.0039}$ | $0.761_{0.017}$ | $\mathbf{0.34_{0.01}}$ | $20.817_{2.523}$ |
| | CP2-PCP | $\mathbf{0.0685_{0.0073}}$ | $\underline{0.0137}_{0.0039}$ | $0.786_{0.017}$ | $0.35_{0.02}$ | $35.426_{4.708}$ |
| | L-CP | $0.0776_{0.0063}$ | $0.0168_{0.0027}$ | $0.775_{0.022}$ | $0.36_{0.02}$ | $\mathbf{0.003_{0.001}}$ |
| | PCP | $0.1024_{0.0079}$ | $0.0249_{0.0047}$ | $0.768_{0.024}$ | $0.35_{0.01}$ | $13.920_{2.513}$ |
| | C-PCP | $\underline{0.0708}_{0.0067}$ | $\mathbf{0.0125_{0.0034}}$ | $0.787_{0.020}$ | $0.35_{0.02}$ | $39.690_{6.311}$ |
| | STDQR | $0.1051_{0.0074}$ | $0.0267_{0.0041}$ | $0.759_{0.018}$ | $0.35_{0.02}$ | $22.148_{3.143}$ |
| | CopulaCPTS | $0.0897_{0.0068}$ | $0.0197_{0.0040}$ | $\underline{0.789}_{0.036}$ | $0.60_{0.79}$ | $18.410_{2.872}$ |
| taxi | Stand. Res. | $0.0067_{0.0029}$ | $\mathbf{0.0000_{0.0002}}$ | $0.862_{0.008}$ | $3.32_{0.05}$ | $\underline{0.004}_{0.001}$ |
| | MSE | $0.0273_{0.0027}$ | $0.0013_{0.0003}$ | $0.814_{0.009}$ | $\mathbf{1.89_{0.01}}$ | $0.172_{0.060}$ |
| | MVCS | $0.0251_{0.0036}$ | $0.0016_{0.0007}$ | $0.827_{0.005}$ | $3.23_{0.07}$ | $0.041_{0.013}$ |
| | HR | $\mathbf{0.0059_{0.0048}}$ | $\mathbf{0.0000_{0.0003}}$ | $0.865_{0.003}$ | $3.48_{0.04}$ | $0.006_{0.001}$ |
| | DR-CP | $0.0151_{0.0040}$ | $0.0005_{0.0004}$ | $0.838_{0.012}$ | $\underline{2.26}_{0.64}$ | $0.007_{0.002}$ |
| | C-HDR | $0.0142_{0.0025}$ | $0.0002_{0.0002}$ | $0.863_{0.007}$ | $2.56_{0.43}$ | $17.261_{4.656}$ |
| | HD-PCP | $0.0980_{0.1509}$ | $0.0295_{0.0535}$ | $0.761_{0.163}$ | $3.60_{1.01}$ | $17.042_{6.860}$ |
| | CP2-PCP | $0.0140_{0.0108}$ | $0.0001_{0.0003}$ | $\mathbf{0.880_{0.018}}$ | $4.34_{1.84}$ | $32.258_{2.918}$ |
| | L-CP | $\underline{0.0061}_{0.0042}$ | $\mathbf{0.0000_{0.0002}}$ | $0.863_{0.009}$ | $3.13_{0.23}$ | $\mathbf{0.002_{0.000}}$ |
| | PCP | $0.0664_{0.1472}$ | $0.0216_{0.0565}$ | $0.789_{0.157}$ | $4.05_{1.56}$ | $14.244_{7.030}$ |
| | C-PCP | $0.1232_{0.1490}$ | $0.0343_{0.0510}$ | $0.741_{0.245}$ | $5.87_{2.65}$ | $36.764_{17.908}$ |
| | STDQR | $0.0518_{0.0943}$ | $0.0098_{0.0218}$ | $0.812_{0.107}$ | $3.23_{0.45}$ | $14.112_{4.665}$ |
| | CopulaCPTS | $0.0091_{0.0056}$ | $\mathbf{0.0000_{0.0002}}$ | $\underline{0.872}_{0.011}$ | $3.81_{1.06}$ | $22.428_{6.120}$ |

