# OpenReview forum: "Conditional Coverage Diagnostics for Conformal Prediction"
_ICML.cc/2026/Conference — ICML 2026 regular_

### Official Review · Reviewer_8AY5 · 2026-02-23

**Soundness:** 4
**Presentation:** 4
**Significance:** 3
**Originality:** 3
**Overall Recommendation:** 4
**Confidence:** 4

**Summary:**

The paper casts conditional coverage estimation as a classification problem. Conditional coverage is violated if and only if any classifier can achieve lower risk than the target coverage. The authors propose a family of metrics that measure this risk, and show experimentally that their proposal works well in comparing competing procedures.

**Compliance With Llm Reviewing Policy:**

Affirmed.

**Key Questions For Authors:**

In using your classification to compare different methods, how sensitive/robust is the resulting decision on the underlying data structure?

**Limitations:**

yes

**Strengths And Weaknesses:**

The authors focus on an important problem which is identifying departures from correct conditional coverage.

As far as I understand, their approach can be used to compare competing methods producing prediction intervals; by doing so, they can choose the best method among the candidate methods. The limitation is that the authors do not directly propose a method that achieves the smallest departures from correct conditional coverage.

---

> ### Author Rebuttal · Authors · 2026-03-30
>
> We sincerely thank the reviewer for the positive review and for recognizing the soundness, presentation, and importance of our approach to identifying departures from correct conditional coverage. Below, we address your noted limitation and your question.
>
> W1. Regarding the limitation that we do not directly propose a method to achieve perfect conditional coverage: while we understand that introducing an evaluation metric naturally raises the expectation for a corresponding correction method, developing such a correction mechanism falls outside the scope of this paper. Modifying predictive sets to strictly guarantee conditional coverage is a distinct and highly complex challenge. Furthermore, we believe that papers on evaluation metrics, like benchmark papers, should not introduce new methods to make them less biased towards specific methods. However, we do take a step in this direction in Section 3.4.: our classifier predicts which coverage is actually achieved by the given sets, so instead of rectifying the sets, we propose to rectify their coverage.
>
>
> Q1. Could you please clarify what you mean by "underlying data structure" in this context? We are currently unsure which specific aspect of the data you are referring to.
>
> We hope these clarifications successfully address your concern in W1. We remain available to discuss Q1 further during the rolling discussion phase.

---

> > ### Author Rebuttal · Reviewer_8AY5 · 2026-04-01
> >
> > By data structure I mean the actual joint distribution of X and Y

---

> > > ### Author Response · Authors · 2026-04-02
> > >
> > > Q1. To clarify, the joint distribution of $(X, Y)$ naturally affects the joint distribution of $(X, Z)$, which is the primary focus of our evaluation (where $Z=\mathbf{1}\\{Y \in C(X)\\}$). While this distribution is partially determined by $(X, Y)$, it also heavily depends on how the set rule $C(X)$ is constructed (i.e., the non-conformity score used). Because we reformulate conditional coverage evaluation as a standard binary classification problem (predicting $Z$ given $X$), all existing machine learning results regarding robustness to different data structures inherently transfer to our setup. In particular, both global robustness and local sensitivity to the underlying data structure is entirely dependent on the choice of the classifier, which is exactly why we conducted the classifier benchmark in Section 4.1.
> > >
> > > To explicitly study the impact of the data structure of $(X,Y)$, we evaluated our method across several real and synthetic examples:
> > > * **Real datasets (Section 4.1):** We benchmarked different classifiers' abilities to identify conditional miscoverage. During the rebuttal, we added four new datasets to the benchmark, confirming a clear trend: performance from standard tabular benchmarks (e.g., Erickson et al., 2025) transfers well to our setting. Table 2 shows that robustness to data structure highly depends on the classifier. Existing metrics (like CovGap) rely on partition-wise estimators that perform poorly in higher dimensions/complex structures. In contrast, using state-of-the-art classifiers (TabICL, TabPFN) yields far more accurate diagnostics. We also include a small experiment on images (Table J.1 in Appendix), where we successfully identify large values of $L_1$-ERT.
> > > * **Synthetic datasets (Section 4.2):** We used a feature space of size $8$ (with $7$ uninformative noise features) where the distribution of $Y|X$ is heteroskedastic. Using the score $S(X,Y)=|Y-f(X)|$, the resulting sets lack conditional coverage, and our method (with a LightGBM classifier) correctly identifies this misbehavior. Conversely, for oracle sets that are perfectly conditional, our method correctly identifies a perfect conditional coverage.
> > > * **New synthetic additions:** We have added two new synthetic experiments during the rebuttal (https://drive.google.com/drive/folders/1vNJcxPxP_ZVr-SobNtgfAr8iQbr69A9o?usp=sharing. The first uses a smoother variance profile where the coverage violation is more subtle. While convergence takes slightly longer, our default LightGBM classifier correctly identifies the violation using only a few samples, whereas partition-wise estimators (used in CovGap) fail entirely. The second new experiment successfully isolates and identifies under- and over-coverage.
> > >
> > > To explicitly answer your question and study the impact of the robustness to the local sensitivity in the distribution of $(X,Y)$, we added one new experiment to study the robustness to small perturbations: we introduced a structural perturbation to the data. We evaluated whether our default diagnostic framework, utilizing the LightGBM classifier, could still accurately identify coverage issues despite this perturbation and the presence of 7 uninformative noise features. As the results demonstrate, our metric successfully detects the conditional coverage perturbation, performing significantly better than partition-wise estimators which struggle under these distorted conditions (note that existing metrics rely on partition-wise estimators).
> > >
> > > In summary, our method is highly robust to the underlying data structure, provided the chosen classifier is robust to that structure. Our framework allows the field to move past weak partition-wise and clustering estimators (less robust to the data structure), unlocking the use of state-of-the-art classifiers (like TabICL or TabPFN) to achieve diagnostic results that are significantly more accurate than previous methods permitted.
> > >
> > > We hope that these clarifications and additional experiments thoroughly address your concerns, and we kindly ask that you reconsider your evaluation of our work in light of these updates.
> > >
> > > Erickson, N., Purucker, L., Tschalzev, A., Holzmüller, D., Desai, P. M., Salinas, D., and Hutter, F. Tabarena: A living benchmark for machine learning on tabular data. In International Conference on Machine Learning, 2025.

---

### Official Review · Reviewer_BAyQ · 2026-03-11

**Soundness:** 3
**Presentation:** 3
**Significance:** 3
**Originality:** 3
**Overall Recommendation:** 5
**Confidence:** 3

**Summary:**

This paper considers the problem of evaluating conditional coverage for conformal prediction methods.
The authors recast the diagnosis of conditional coverage violations as a classification task: given a suitable loss function $\ell$, conditional coverage is violated if there exists a classifier that achieves lower risk than the constant classifier corresponding to the target coverage level.
Based on this idea, the excess risk of target coverage ($\ell$-ERT) metric is defined as the gap between these two risks, which quantifies deviations from conditional coverage.
The paper instantiates $\ell$-ERT for three choices of loss $\ell$ and proposes to estimate it by training a classifier on held-out data.
Empirically, the authors benchmark common classifiers for estimating $\ell$-ERT, and show on a simple synthetic example that the proposed diagnostics more reliably identify the presence/absence of conditional coverage than existing baselines. The supplementary material further applies $\ell$-ERT to benchmark existing conformal prediction methods.

**Compliance With Llm Reviewing Policy:**

Affirmed.

**Final Justification:**

With their rebuttal, the authors commited to add details that improve the clarity of their manuscript, and provided additional content that address my initial concerns about experimental evaluation.

**Key Questions For Authors:**

Questions:
1. Could the authors clarify the default classifier used in Section 4.2, and comment on sensitivity of the conclusions there to the choice of classifier?
2. The experiment in Section 4.2 compares two extreme cases: an oracle procedure with exact conditional coverage and a standard method with only marginal coverage. Could the authors include an additional synthetic setting with a more intermediate form of conditional-coverage violation? This could help illustrate empirically whether $\ell$-ERT tracks the severity of violations rather than only distinguishing the two endpoints.
3. From Section 5, "While the reliability of our metrics naturally depends on the quality of the learned classifier this dependency is also a strength." Could the authors expand on why this is a strength?

**Limitations:**

yes

**Strengths And Weaknesses:**

Strengths:
- The paper is well written and easy to follow
- As there is significant ongoing research into developing conformal prediction methods with approximate conditional coverage guarantees, methods to reliably evaluate them are valuable
- The benchmarking of existing conformal prediction methods according to the proposed metric, as well as the release of an open-source package to facilitate evaluation of future methods, is helpful for future research

Weaknesses:
- While informative, the synthetic example in Section 4.2 to compare against existing metrics is quite simple
- Although the paper studies classifier choice in Section 4.1, the practical sensitivity of the diagnostic to the choice and training of $h$ remains non-trivial

---

> ### Author Rebuttal · Authors · 2026-03-30
>
> We sincerely thank the reviewer for the positive assessment, particularly for highlighting the clarity of our writing, the importance of evaluating approximate conditional coverage, and the value of our open-source package for future research. Below, we address your weaknesses and questions.
>
> W1. We acknowledge that the synthetic setup in Section 4.2 is relatively simple. However, we believe this actually strengthens our central argument: the fact that existing metrics fail even on such a *simple* example is a strong signal of their inherent limitations. Furthermore, evaluating these existing baseline metrics requires knowing the ground-truth conditional coverage, which inherently restricts the complexity of the synthetic tests we can design. Please also note that we can extend our comparison to real datasets in Table 2, where CovGap is evaluated using its underlying partition-wise classifier, performing significantly more poorly than other classifiers. Still, we have added a more complex experiment (cf Q2).
>
> W2. We agree that the practical sensitivity to the choice and training of the classifier is non-trivial, which is exactly why we dedicated Section 4.1 to benchmarking them. However, it is important to recognize that existing metrics (like CovGap) inherently rely on partition-wise estimators, which perform significantly worse than modern state-of-the-art classifiers. Therefore, we view this dependence not as a flaw, but as a mechanism that allows our diagnostic to leverage powerful, modern classification tools rather than remaining bottlenecked by basic clustering strategies.
>
> Q1. The default classifier used in Section 4.2 is LightGBM. We chose it because it emerged as the recommended default from our classifier benchmark in Section 4.1. We have now explicitly stated this in the main text, thank you for pointing out this omission. Regarding sensitivity, the conclusions in Section 4.2 remain highly consistent across all other classifiers evaluated in Section 4.1, with the sole exception of the partition-wise classifier, which unsurprisingly fails in this specific experiment due to the feature dimension (of only 8).
>
> Q2. Thank you for this excellent suggestion. To better illustrate how the metric tracks the severity of violations rather than just distinguishing the endpoints, we have added new experiments that you can find here: https://drive.google.com/drive/folders/1vNJcxPxP_ZVr-SobNtgfAr8iQbr69A9o?usp=sharing
>
> * **Smoother variance:** The setup is identical to the one in Section 4.2 (Figure 3), but the variance perturbation is much smoother. As a result, the conditional coverage violation is more subtle, particularly due to the inclusion of 7 uninformative noise features. We observe that our default classifier (LightGBM) yields a significantly better diagnostic than the partition-wise classifier (which corresponds to the estimator used in CovGap). While convergence to the true value takes slightly longer in this smoothed setting, the LightGBM classifier correctly identifies the conditional coverage violation using only a few samples. In contrast, the partition-wise estimator fails to provide an accurate diagnostic even with 100,000 samples.
> * **Evaluating over- and under-coverage:** Using the setup from section 4.2 Figure 3, we have added the estimation for L1 over- and under- coverage. For the left plot (Standard CP), not surprisingly, the over- and under-coverage are almost equal as the marginal coverage constraint ensures this. Due to randomness in the calibration data, the marginal coverage is not exactly $0.9$ that is why both quantities are not exactly equal. For the conditional (right plot, where perfect conditional coverage equals $0.9$), we test how far the set is from a target coverage of $0.8$. The $L_1$-ERT for under-coverage correctly outputs $0.0$ (identifying that the strategy is never under-confident), while the $L_1$-ERT for over-coverage outputs $0.1$, exactly matching the theoretical $0.9 - 0.8$ over-coverage.
>
>
> Q3. We argue this dependency is a strength because it allows the evaluation framework to automatically improve alongside ongoing advancements in the machine learning classification literature, and leverage different classifiers suited to different data types like images or text. We will make sure to clarify this in the revised text. As touched upon in W2, existing metrics like CovGap estimate the exact same theoretical quantity as our $L_1$-ERT (specifically, $\mathbb{E}[|\mathbb{P}(Y\in C(X)|X) - (1-\alpha)|]$), but they are structurally constrained to use weak partition-wise estimators. By framing this as a standard classification task, we achieve a far more accurate estimation of this exact same quantity simply by utilizing a stronger classifier.
>
> We hope to have answered the reviewer’s concerns. We are happy to provide further clarifications should the reviewer have additional questions.

---

> > ### Author Rebuttal · Reviewer_BAyQ · 2026-04-02
> >
> > Thank you for the thorough response, and especially for the new experimental results, which I believe constitute useful additions to the current manuscript. I will revise my score accordingly.

---

### Official Review · Reviewer_95nw · 2026-03-15

**Soundness:** 3
**Presentation:** 3
**Significance:** 2
**Originality:** 3
**Overall Recommendation:** 4
**Confidence:** 4

**Summary:**

This paper proposes a novel evaluation framework for approximating conditional coverage performance in conformal prediction. Specifically, it formulates conditional coverage estimation as a binary classification problem and introduces the metric ‘excess risk of the target coverage (ERT)’, which measures how far the true feature-conditional coverage deviates from the target level $1-\alpha$. Since the true conditional coverage is unknown, the paper estimates a lower bound on ERT using an empirical binary classifier. The authors demonstrate the effectiveness of the proposed approach through synthetic experiments and apply it to a range of real-world datasets.

**Compliance With Llm Reviewing Policy:**

Affirmed.

**Final Justification:**

I maintain a weak accept on this paper after taking the authors’ rebuttal into account.

**Key Questions For Authors:**

For the synthetic experiments, could you share the estimated overcoverage and undercoverage for the non-conditional method using the proposed metric? It would be interesting to see them as we vary $X_1$

**Limitations:**

yes

**Strengths And Weaknesses:**

Strengths:
This paper is well structured and well organized; it is easy to follow overall.
The proposed method is well motivated by the well-recognized challenge of estimating feature conditional coverage in the conformal prediction literature, and it contributes meaningfully to this gap.

Weaknesses:
The main concern I have about the proposed method is its practical limitation. Computing this metric involves training additional classifiers, which imposes extra computational burden and data costs. Additionally, the reliability of the metric depends on the quality of this auxiliary classifier. As the authors already demonstrated in Table 2, different classifier choices and loss functions $\ell$ can lead to very different $\ell-ERT$ values.  Although the authors acknowledge the compute costs and provide recommendations regarding the choice of $\ell$ and classifier, in practice it may remain unclear how well these guidelines generalize to a broader range of datasets or tasks.

---

> ### Author Rebuttal · Authors · 2026-03-30
>
> We sincerely thank the reviewer for the encouraging feedback and for acknowledging our paper's clear structure, organization, and motivation in addressing the critical challenge of evaluating feature-conditional coverage. Below, we address your concerns regarding practical limitations and provide the additional experimental details you requested.
>
> W1.
> Regarding the "extra data costs," our method actually does *not* require more data. It utilizes the exact same hold-out test set used for traditional evaluation. In fact, because modern classifiers generalize better, our method often requires *less* data to achieve a reliable estimate compared to existing metrics. (If data is exceptionally scarce, one could technically use the calibration data, though we do not recommend it).
>
> Regarding the practical burden of calculating the metric: that is why we release a package to calculate the ERT metrics efficiently:
> ``` python
> from anonymous_package import ERT
>
> ERT_value = ERT().evaluate(X, Z, alpha) # Z = 1_{Y \in C(X)}
> ```
>
>
> We understand your concerns regarding the reliance on an auxiliary classifier. However, we would like to gently point out that existing methods, such as CovGap, inherently suffer from the exact same dependency: they just rely on partition-wise classifiers (for which there are multiple choices, e.g. the number of groups), which, as we demonstrate in Table 2, perform significantly worse. A major advantage of our formulation is that the ERT metric will automatically improve simply by leveraging future advancements in the classifier literature. Additionally, our formulation can leverage different classifiers for different data types, such as images or language. The alternative is to bake an implicit classifier into the metric, meaning that the metric can’t easily be adapted. Furthermore, the other metric WSC also suffers from the computational cost of its estimation, as it takes on average 11.6 seconds to compute per 1,000 samples, which is 5x the runtime of ERT with our suggested classifier (we have added this number in Table 2, see below).
>
> Regarding the concern about generalization, we followed your implicit suggestion and expanded our evaluation by adding 4 new datasets (bringing the total to 8, including one datasets with 1025 features). The trend remains clear: performance from standard tabular benchmarks (e.g., Erickson et al., 2025) transfers very well to our setting. Therefore, we can reliably lean on these existing benchmarks to guide classifier selection. The new table is available at https://drive.google.com/drive/folders/1vNJcxPxP_ZVr-SobNtgfAr8iQbr69A9o?usp=sharing
>
> Finally, regarding the choice of the loss function $\ell$, we discuss this in Table 1: the choice depends entirely on the theoretical quantity you wish to estimate. We highly recommend using the $L_1$-ERT as the default. Because it is not a strictly proper score, the loss is often easier to minimize empirically, and it provides a very straightforward geometric interpretation: it directly estimates $\mathbb{E}[|\mathbb{P}(Y\in C(X)|X)-(1-\alpha)|]$.
>
> Q1. Following your suggestion, we have added a dedicated experiment specifically isolating over- and under-coverage. Using the setup from section 4.2 Figure 3, we have added the estimation for L1 over- and under- coverage. For the left plot (Standard CP), not surprisingly, the over- and under-coverage are almost equal as the marginal coverage constraint ensures this. Due to randomness in the calibration data, the marginal coverage is not exactly $0.9$, which is why both quantities are not exactly equal. For the conditional (right plot, where perfect conditional coverage equals $0.9$), we test how far the set is from a target coverage of $0.8$. The $L_1$-ERT for under-coverage correctly outputs $0.0$ (identifying that the strategy is never under-confident), while the $L_1$-ERT for over-coverage outputs $0.1$, exactly matching the theoretical $0.9 - 0.8$ over-coverage. (https://drive.google.com/drive/folders/1vNJcxPxP_ZVr-SobNtgfAr8iQbr69A9o?usp=sharing)
>
> We hope the new experiments and clarifications provided above successfully resolve the concerns raised. We remain available for any further discussion.
>
>
> Erickson, N., Purucker, L., Tschalzev, A., Holzmüller, D., Desai, P. M., Salinas, D., and Hutter, F. Tabarena: A living benchmark for machine learning on tabular data. In International Conference on Machine Learning, 2025.

---

> > ### Author Rebuttal · Reviewer_95nw · 2026-04-04
> >
> > I thank the authors for their thorough clarifications and additional numerical results. My main concerns have been satisfactorily addressed, and I am pleased to maintain my positive stance and current score.

---

### Official Review · Reviewer_xEH2 · 2026-03-19

**Soundness:** 2
**Presentation:** 3
**Significance:** 2
**Originality:** 3
**Overall Recommendation:** 4
**Confidence:** 4

**Summary:**

This paper aims to improve the evaluation of conditional coverage of conformal predictors. There are many existing metrics for this, but they have their shortcomings.

The paper casts the problem of conditional coverage estimation as one of binary classification. Hence, modern-day highly accurate classifiers can estimate the risk associated with under (and over) conditional coverage. This risk can take on many forms, including the Bregman divergence of convex functions.

Experimental results on synthetic and real-world datasets illustrate the detection of (mis)coverage using the proposed method, compared to existing methods.

**Compliance With Llm Reviewing Policy:**

Affirmed.

**Final Justification:**

The paper evaluates conditional coverage for conformal predictors. The authors provided many clarifications during the rebuttal period; I strongly believe these should be incorporated in the final version of the paper. As a result, I am sticking with my evaluation of a "weak accept".

**Key Questions For Authors:**

Refer to the Strengths and Weaknesses section.

**Limitations:**

The provided impact statement is not informative. Please update it.

**Strengths And Weaknesses:**

__Strengths__

1. The paper is well motivated to evaluate conditional coverage for conformal predictors.
2. The proposed method is presented well and is easy to follow. I personally also liked the generalization to the Bregman divergence of convex functions (Section 3.2) and the separate over- and under-coverages (Section 3.3).

__Weaknesses__

1. $\ell\text{-ERT}(h) \leq \ell\text{-ERT}$ is a very useful property, and is utilized in the paper. Is there proof for this claim? I did not see one.
2. Table 3 suggests that the proposed $L_{1}$-ERT and $L_{2}$-ERT are accurate diagnostics. Although their values do not change much between the conditional and non-conditional settings, the change demonstrates the correct trend. Similarly, CovGap, WCovGap, and EOC also display the correct trends. Why are they deemed failures?
3. The proposed metric is dependent on the predictor $h$ used for the estimation. While providing flexibility, it makes standardization of the evaluation difficult. Any thoughts in this direction?

__Suggestions and Clarity__

1. I suggest moving Fig. 1 to the experiments section. Also, it might be worthwhile to remind the reader that CovGap and $L_{1}$-ERT estimate the same theoretical quantity, and are hence in the same plot.
2. "WSC tends to provide a pessimist[ic] estimate of the conditional coverage violation by overfitting the test set by isolating miscovered points." (lines 156-160, column 2). Could this be explained more?
3. Are there any experimental results to complement the generalizations provided in Sections 3.2 and 3.3?
4. Could the metric reported in Table 2 be explained more clearly?

---

> ### Author Rebuttal · Authors · 2026-03-30
>
> We sincerely thank the reviewer for the constructive feedback and for highlighting the clear motivation and presentation of our proposed method, as well as the generalizations in Sections 3.2 and 3.3. Below, we address your questions and suggestions in detail.
>
> W1. This statement is a direct consequence of the fact that $R(h) \geq \textrm{inf}_{h:\mathcal{X}\to[0, 1]} R(h)$ for all $h:\mathcal{X}\to[0, 1]$. We will mention it in the paper.
>
> W2. It is true that comparing these metrics across different setups reveals the correct trend. However, in practice, a practitioner only observes the metric's absolute value in isolation, not a comparative trend. For example, observing a CovGap of $0.016$ might falsely lead one to conclude that the deviation $\mathbb{E}[|\mathbb{P}(Y\in C(X)|X)-(1-\alpha)|]$ is very low. In contrast, our $L_1$-ERT correctly identifies that this deviation is at least $0.091$. A similar argument applies to WSC, which yields $0.79$ in a setting where the true coverage is exactly $0.90$ so the WSC should be $0.90$. We now state clearly in the paper what those ticks represent.
>
> W3. Indeed the metric depends on the predictor. To aid standardization, we performed a classifier benchmark (Section 4) and explicitly suggest using LightGBM as the default classifier (line 365, p. 7 column 1), which we also make the default in our package. Hence, users that want to use a justifiable choice of the predictor can rely on our default. However, we believe that it is useful to empower the user to use different predictors, for example, for other modalities like images where other predictors are better suited to the data. Additionally, some popular metrics like CovGap also rely on partition-wise estimators, which also have a dependency on a choice of partitions, and may be more sensitive to this choice because the underlying histogram-based classifier is weak.
>
> S1. Thank you for the suggestion! We agree that reminding the reader that both CovGap and L1-ERT estimate the same theoretical quantity will make the plot much easier to understand. We will update the text and caption accordingly.
>
> S2. We apologize for the lack of clarity. Please note that the first part of that sentence specifies the setting "under conditional coverage" (meaning the method *does* achieve a true conditional coverage of $1-\alpha$). When this holds true, WSC tends to provide a pessimistic estimate (below $1-\alpha$), which we observe in Table 3. To build an intuition: imagine drawing $1,000$ samples from a Bernoulli distribution $\mathcal{B}(1-\alpha)$. WSC with $\delta=0.1$ looks for the subset of $100$ samples with the lowest coverage. Due to pure randomness, this "worst slab" will naturally fall below $1-\alpha$. This overfitting effect worsens in higher dimensions, where the metric can more easily find arbitrary feature-space slabs that artificially group the miscovered points ($0$s) together.
>
> S3.
>  * Regarding Section 3.2, the $L_1$-ERT is a direct application of this section. We will mention it in the paper. Since we propose $L_1$-ERT as the default metric, most of the experiments in the paper serve as experimental validations of Section 3.2.
>
>  * Regarding Section 3.3, we utilize it to evaluate the KL-ERT in a classification task in Appendix J.2. Following your recommendation, we have also added a new experiment (see https://drive.google.com/drive/folders/1vNJcxPxP_ZVr-SobNtgfAr8iQbr69A9o?usp=sharing). Using the setup from section 4.2 Figure 3, we have added the estimation for L1 over- and under- coverage. For the left plot (Standard CP), not surprisingly, the over- and under-coverage are almost equal as the marginal coverage constraint ensures this. Due to randomness in the calibration data, the marginal coverage is not exactly $0.9$, which is why both quantities are not exactly equal. For the conditional (right plot, where perfect conditional coverage equals $0.9$), we test how far the set is from a target coverage of $0.8$. The $L_1$-ERT for under-coverage correctly outputs $0.0$ (identifying that the strategy is never under-confident), while the $L_1$-ERT for over-coverage outputs $0.1$, exactly matching the theoretical $0.9 - 0.8$ over-coverage.
>
> S4. To clarify the metric reported in Table 2: referring to Figure I.1, the highest diagnostic value achieved by any classifier is $0.08$. For each classifier, we divide the values obtained at each test sample size by $0.08$ and multiply by $100$%. This gives us the average percentage of the "best" ERT diagnostic for that specific dataset. These percentages are then averaged across all datasets to produce the final reported metrics. We will ensure this normalization process is explained more clearly in the revised text.
>
> We thank the reviewer for their insightful comments. We hope the new experiments and clarifications provided above successfully resolve the concerns raised. We remain available for any further discussion.

---

> > ### Author Rebuttal · Reviewer_xEH2 · 2026-04-03
> >
> > Thank you for your rebuttal.
> >
> > 1. __[W2]__ I do not follow this justification. 0.009 (from $L_{1}$-ERT) was high enough to demonstrate deviation (i.e., ticked), but 0.016 (from CovGap and WCovGap) was not (i.e., crossed)? I would, in fact, argue that both 0.009 and 0.091, returned by $L_{2}$-ERT and $L_{1}$-ERT, are very low absolute values.
> > 2. __[W3]__ Elaborating more on the standardization would be helpful for the paper. The experiments aid standardization, yes, but only for tabular data and for the current choice of models (and optimizers).
> > 3. __[S2]__ Is this pessimism of WSC proven? Is this observed experimentally by you only or by others as well? A statement like this in a research paper needs to be clarified and supported.
> > 4. __[S3]__ The additional experiments, along with the visualizations, are very helpful.

---

> > > ### Author Response · Authors · 2026-04-05
> > >
> > > **W2.** This is due to the quantities that the methods estimate (see Table 1), but we agree that this can be misunderstood and provide an updated table with more explanations in the google drive link.
> > > * The L1-ERT of $0.091$ estimates the expected absolute deviation from 90% coverage. It means that on average, the coverage is 9.1% away from 90%, which is quite large, and close to the true value of $0.098$ from Figure 1. It is much larger than the 1.6% estimated by CovGap. Moreover, CovGap also estimates 1.4% in the conditional case, meaning that 1.6% is not enough to accurately diagnose miscoverage, while L1-ERT estimates are closer to zero in the conditional case.
> > > * The L2-ERT of $0.009$ is smaller because it estimates the expected *squared* deviation from 90% coverage, that is, $E_X[(P(Y \in C(X)|X) - (1-\alpha))^2]$. The squared distance makes the ground truth smaller, as in general $P(Y \in C(X)|X) \approx 1-\alpha$. In the example of Table 3, the true value is $0.012$ (according to Figure 1). To bring it to the same scale, we could compute its square root (as in RMSE vs MSE) and obtain $\sqrt{0.009} \approx 0.094$, similar to the L1 estimate. Moreover, we note that despite the smaller scale, the $0.009$ in the non-conditional case can be clearly distinguished from the $0.000$ in the conditional case: the scale to interpret the deviation is just different for the L2-ERT, but in squared distance, $0.009$ is quite large
> > >
> > >
> > > **W3.** We will elaborate more in the paper. While the primary focus of our analysis is indeed mainly on tabular data, we do have experiments on images in Appendix Table J.1. In general, each modality (tabular data, images, text, graphs, etc.) has its own state-of-the-art classifiers. As we successfully recast the problem as a binary classification one, our overall recommendation is simply to take a strong classifier suited for the relative data domain. Our tabular experiments support that classifier rankings from general benchmarks transfer to ERT. We will add a paragraph at the end of the section “Comparing different classifiers” to provide further details.
> > >
> > > However, performing additional benchmarks on different modalities is beyond the scope of the paper. Existing benchmarks already exist for each modality, so we will instead refer to those papers in the main text. A non-exhaustive list includes: ImageNet for images, Prakash et al. (2023) for graphs, and Wang et al. (2024) or Wang et al. (2018) for text.
> > >
> > > **S2.** Completely agreed, this statement will be backed up through additional information. We will make sure to add more detailed explanations in the paper.
> > >
> > > The basic reason for why WSC underestimates the true WSC is the same as for why the train loss underestimates the test loss in machine learning: overfitting. It can be proven as follows: Let $S$ be the set of allowed slabs, and let $C_s$ be the coverage on slab $s \in S$. Then, for each $s$, $E[\inf_{s’ \in S} C_{s’}] \le E[C_s]$ (taking the expectation over draws of the dataset). Since it holds for all $s$, we also have $E[\inf_{s’ \in S} C_{s’}] \le \inf_{s \in S} E[C_s]$ (and even $<$ unless the inf is always taken by the same $s$). Under conditional coverage, $E[C_s] = 1-\alpha$, and WSC underestimates $1-\alpha$.
> > >
> > >
> > > We have added a new plot (in https://drive.google.com/drive/folders/1vNJcxPxP_ZVr-SobNtgfAr8iQbr69A9o?usp=sharing  ) showing that WSC indeed decreases with $d$ under perfect coverage. This reflects the behavior of a lower bound in the original WSC paper (https://arxiv.org/pdf/2004.10181v3, p. 15, equation below Eq. (24)) that also decreases with $d$:
> > > $$\inf \mathrm{WSC}_n(\widehat C, v) \ge 1 - \alpha - O(1)\sqrt{\frac{\min\{d, \log |V|\} \log n}{\delta n}}.$$
> > >
> > > For the experiment, we drew $n$ i.i.d. samples of $(X,Z)$ where $X \sim \mathcal{U}([0, 1]^d)$ and $Z \sim \mathcal{B}(1-\alpha)$ is independent of $X$ (thereby mimicking a conditional coverage setup), and we plotted the average estimated WSC as a function of the input dimension. We repeated this experiment with varying numbers of samples. We attached the result with 1,000 slabs, and we checked that different numbers of slabs yield similar results. Our results explicitly corroborate our statement that WSC is a pessimistic metric under conditional coverage.
> > >
> > > We hope to have fully answered the reviewer’s concerns.
> > >
> > > Yubo Wang et al. MMLU-Pro: A more robust and challenging multi-task language understanding benchmark. CoRR, abs/2406.01574, 2024.
> > >
> > > Olga Russakovsky et al. Imagenet large scale visual recognition challenge. IJCV, 2015.
> > >
> > > Vijay Prakash Dwivedi et al. Benchmarking graph neural networks. Journal of Machine Learning Research, 24(43):1–48, 2023.
> > >
> > > A. Wanget al. “Glue: A multi-task benchmark and analysis platform for natural language understanding,” 2018 EMNLP workshop
> > >
> > > Cauchois, M. et al (2021). Knowing what you know: valid and validated confidence sets in multiclass and multi label prediction. Journal of Machine Learning Research 22(81), 1–42.

---

### Decision · Program_Chairs · 2026-04-30

**Decision:**

Accept (regular)

**Comment:**

The paper considers the problem of improving the evaluation aspects of conditional coverage in conformal prediction area. The key idea behind the paper is to formulate conditional coverage estimation as a binary classification problem. This allows us to leverage existing work on classifiers to estimate the risk associated with under/over conditional coverage where multiple risk functions could be used including Bregman divergence of convex functions. Experimental results on synthetic and real-world data demonstrate the efficacy of the proposed method over baselines.

There is a consensus among all reviewers' after discussion on weak accept for the paper. I concur with them and strongly encourage the authors to incorporate all the review comments / discussion to improve the final paper to make it useful for the community.